# Transcriptome profiling of tendon fibroblasts at the onset of embryonic muscle contraction reveals novel force-responsive genes

**Pavan K Nayak[†], Arul Subramanian[†], Thomas F Schilling***

Department of Developmental and Cell Biology, University of California, Irvine, United States

**Abstract** Mechanical forces play a critical role in tendon development and function, influencing cell behavior through mechanotransduction signaling pathways and subsequent extracellular matrix (ECM) remodeling. Here, we investigate the molecular mechanisms by which tenocytes in developing zebrafish embryos respond to muscle contraction forces during the onset of swimming and cranial muscle activity. Using genome-wide bulk RNA sequencing of FAC-sorted tenocytes we identify novel tenocyte markers and genes involved in tendon mechanotransduction. Embryonic tendons show dramatic changes in expression of *matrix remodeling associated 5b* (*mxra5b*), *matrilin 1* (*matn1*), and the transcription factor *kruppel-like factor 2a* (*klf2a*), as muscles start to contract. Using embryos paralyzed either by loss of muscle contractility or neuromuscular stimulation we confirm that muscle contractile forces influence the spatial and temporal expression patterns of all three genes. Quantification of these gene expression changes across tenocytes at multiple tendon entheses and myotendinous junctions reveals that their responses depend on force intensity, duration, and tissue stiffness. These force-dependent feedback mechanisms in tendons, particularly in the ECM, have important implications for improved treatments of tendon injuries and atrophy.

**\*For correspondence:**
tschilli@uci.edu

[†]These authors contributed equally to this work

**Competing interest:** The authors declare that no competing interests exist.

## Editor's evaluation

This valuable manuscript presents solid evidence that identifies potential force-responsive gene expression responses within tenocytes and developing tendons by comparing unperturbed animals to those with paralyzed muscles. A handful of these force-responsive genes are then validated, which reveals that force-responsive gene expression differs between individual tendons or local biophysical environments and shows a phenotype in mutants for a force-responsive gene expressed during tendon development. Future work that further explores how these particular examples relate to broader force-responsive gene expression programs and that identifies stronger phenotypes when force-responsive gene expression is disrupted will strengthen its conclusions. This work is of interest to the fields of developmental biology, mechanobiology, muscle and tendon biology.

## Introduction

All cells experience mechanical forces from their environments, from adhesive interactions between adjacent epithelial cells to structural interactions with the surrounding extracellular matrix (ECM). A key question is how cells adapt and respond to force by modifying their local microenvironment. Force-responsive cellular mechanisms have been implicated in cell differentiation (*D'Angelo et al., 2011*), morphogenesis (*Hamada, 2015*; *Keller et al., 2008*), tissue maintenance and repair (*Riley*

*et al., 2022*; *Zhang et al., 2022*). However, these mechanisms remain understudied in vivo, particularly those that involve cell–ECM interactions. Dramatic examples include tendons and ligaments of the musculoskeletal system. Tendons experience a broad range of contractile forces from muscles, such as extreme stretching forces on the human Achilles tendon during exercise, and their constitutive fibroblast populations (called tenocytes) constantly remodel the surrounding ECM to adapt (*Subramanian and Schilling, 2015*; *Wang, 2006*). Tendon injuries and atrophy with aging are very common, and a better understanding of the roles of force in tendon development will aid in developing effective treatments.

Tendons are ECM-rich structures that connect muscles to cartilages and bones as well as to softer tissues. The events leading to the proper formation of their attachments relies largely upon cell–ECM interactions (*Schweitzer et al., 2010*; *Subramanian and Schilling, 2015*). For example, in the embryonic zebrafish trunk, myotendinous junctions (MTJs) at the vertical myosepta (VMS) of developing somites form via distinct tendon-independent and -dependent stages (*Subramanian and Schilling, 2015*). In the tendon-independent phase, myofibers differentiate and secrete ECM proteins such as Thbs4b that localize to the pre-tendon ECM and mediate initial fiber attachment. This coincides with tendon progenitor cell (TPC) migration into the MTJ. Later, in response to muscle contraction, TPCs differentiate into mature tenocytes and extend long microtubule-rich processes laterally into the surrounding ECM of the VMS, with which they regulate ECM composition locally in response to force (*McNeilly et al., 1996*; *Pingel et al., 2014*; *Subramanian et al., 2018*). Contractile forces acting on these MTJs activate transforming growth factor β (TGF-β) signaling in TPCs (*Berthet et al., 2013*; *Pryce et al., 2009*; *Subramanian et al., 2018*). Although not necessary for TPC specification, TGF-β induces expression of the transcription factors Scleraxis (Scx) and Mohawk (Mkx), which drive tenocyte fate by directly promoting transcription of collagens (i.e. Col1a1, Col1a2, Col12a1, and Col14) enriched in tendon ECM (*Berthet et al., 2013*; *Maeda et al., 2011*).

Cell type and ECM composition differ along the length of many tendons to aid in load bearing and force transmission. For example, the enthesis region where a tendon attaches to cartilage or bone is structurally graded in stiffness with fibrocartilage closer to the bone. This helps buffer mechanical stress between the elastic tendon tissue and rigid bony matrix (*Lu and Thomopoulos, 2013*). Fibrocartilage cells co-express Scx and Sox9, both direct transcriptional regulators of collagens, and muscle activity regulates the ratio of their expression levels (*Blitz et al., 2013*; *Subramanian et al., 2023*; *Zelzer et al., 2014*). This changes collagen levels, fibril size, and organization during injury or repair, as has been shown both in vitro and ex vivo (*Ireland et al., 2001*; *Pingel et al., 2014*). We have also shown that muscle contraction is required for embryonic tenocyte maturation, morphogenesis and ECM production in zebrafish tendons in vivo (*Subramanian et al., 2018*; *Subramanian and Schilling, 2014*).

To identify genes regulated by muscle contraction in tendons we have performed genome-wide bulk RNA-sequencing (RNA-seq) on FAC-sorted tenocytes of zebrafish embryos during the onset of muscle contractions and active swimming behavior. In addition to upregulation of known tenocyte markers, we find several other genes up- or downregulated as tendons differentiate that have not been implicated in tenocyte development or mechanotransduction. These include genes encoding two ECM proteins, Matrix Remodeling Associated 5b (*mxra5b*) and Matrilin 1 (*matn1*), as well as the transcription factor Kruppel-like factor 2a (*klf2a*). We confirm that muscle contraction regulates their transcription in tenocytes at later stages, after the onset of cranial muscle activity, by comparing wild-type and paralyzed embryos. Using genetic and physiological perturbations of muscle contraction in vivo, we show gene expression changes both in whole embryos and sorted tenocytes. Quantitative in situ methods show that their expression is contained within embryonic tendon entheses and MTJs and that their transcriptional responses to force vary depending on the strength and continuity of muscle contraction. These findings provide insights into tendon attachment specific and force-dependent feedback mechanisms in tendons during development in vivo, which have important implications for improved treatments for tendon disease, injury, and atrophy.

# Results

## Onset of active muscle contraction alters tenocyte gene expression

We previously showed that trunk tenocytes in zebrafish undergo dramatic morphological transformations when muscle contractions begin (*Subramanian et al., 2018*; *Subramanian et al., 2023*). These occur when embryos transition from twitching (36 hr post-fertilization, hpf) to free-swimming behaviors (48 hpf), as well as between sporadic jaw contractions at 60 hpf, and free-feeding behavior at 72 hpf (*Figure 1A–D*). These morphological changes likely reflect force-induced transcriptional changes in tenocytes, in addition to changes driving differentiation. To identify potential force-responsive factors, we conducted RNA-seq with FAC-sorted populations of *Tg(scxa:mCherry)*-positive tenocytes isolated from dissociated twitching (36 hpf) or free-swimming embryos (48 hpf). The *Tg(scxa:mCherry)* line predominantly labels both embryonic trunk and cranial tenocytes. We FACS-sorted mCherry+ cells using WT stage-matched non-fluorescent embryos as negative controls (*Figure 1—figure supplement 1*). Differential expression analysis revealed 2788 differentially expressed genes (DEGs) between twitching and free-swimming stages with p-value <0.05 (*Figure 1E*; *Supplementary file 1*). These included known tenocyte markers such as *tnmd*, *mkxa,* and *egr1* upregulated at swimming (*Figure 1E, F*), confirming that many of the sorted mCherry+ cells were tenocytes or TPCs. Principle components associated with biological replicates segregated according to experimental condition (36 vs. 48 hpf), validating library preparation (*Figure 1G*; *Figure 1—figure supplement 1*). GO analysis for Biological Process terms associated with the top DEGs showed significant enrichment for 'skeletal system development' and 'ECM organization' (*Figure 1H*). Surprisingly, these included *col2a1a* and *col9a1a*, which are typically associated with cartilage development and morphogenesis (*Figure 1F*) suggesting that an early subset of *scxa+* cells in embryonic tendons are specified as developing enthesis cells (*Subramanian et al., 2023*). Dual-expressing *scxa/sox9a+* cells localize to cartilage attachment sites of cranial muscles at 48 hpf, prior to the onset of jaw movements (*Figure 1A–D*), consistent with specification of enthesis progenitors before the tendons or their skeletal muscle attachments become functional. These results also fit with recent single-cell sequencing studies of enthesis lineage trajectories in mice (*Fang et al., 2022*).

To identify cell signaling pathways implicated in force responses during embryonic tendon development, we analyzed our DEG list using ShinyGO (*Ge et al., 2020*; *Supplementary file 2*) and DAVID (*Supplementary file 3*), both of which interrogate Gene Ontology and KEGG pathway databases (*Huang et al., 2009*). ShinyGO identified DEGs associated with 52 different pathways with FDR <0.05, including TGF-β, MAPK, Wnt, and Notch signaling, along with cell–cell adhesion and cell–ECM adhesion (*Supplementary file 2*). DAVID identified many of the same pathways as well as DEGs involved in RA metabolism, an emerging pathway of interest in tendon development (*McGurk et al., 2017*; *Supplementary file 3*).

Because our RNA-seq datasets were obtained from tenocytes during the onset of muscle contractions and swimming we also searched for DEGs associated with mechanosensitive pathways. Three genes of particular interest, *matn1*, *klf2a*, and *mxra5b*, stood out based on their force-dependent regulation in other biological contexts or regulation by TGF-β, a well-known force-responsive signal (*Maeda et al., 2011*; *Subramanian and Schilling, 2015*). The top-most upregulated gene was *matn1*, which encodes an ECM protein highly enriched in cartilage; Matn1 enhances chondrogenesis of synovial fibroblasts treated with TGF-β (*Pei et al., 2008*). The transcription factor *klf2a* was also strongly upregulated; Klf2 and Klf4 have been implicated in enthesis development in mammalian tendons. Klf proteins also repress TGF-β signaling in endothelial cells (*Boon et al., 2007*; *Li et al., 2021*) and *klf2a* expression is mechanosensitive during heart valve development (*Steed et al., 2016*). The third DEG of particular interest was *mxra5b,* which encodes an ECM protein expressed in both tendons and ligaments during chick development (*Robins and Capehart, 2018*) and regulated by TGF-β in cultured human kidney epithelial cells (*Poveda et al., 2017*). Though other potentially mechanosensitive genes were present in our bulk RNA-seq dataset, we focused on *matn1*, *klf2a*, and *mxra5b* for further analysis based on evidence implicating them in mechanotransduction in other tissue contexts.

## *matn1*, *klf2a*, and *mxra5b* are expressed in cranial and trunk tenocytes in vivo

To verify tenocyte-specific expression of *matn1*, *klf2a* and *mxra5b*, we performed in situ hybridization (ISH). Conventional chromogenic ISH for *matn1* detected no expression at 36 hpf but very strong

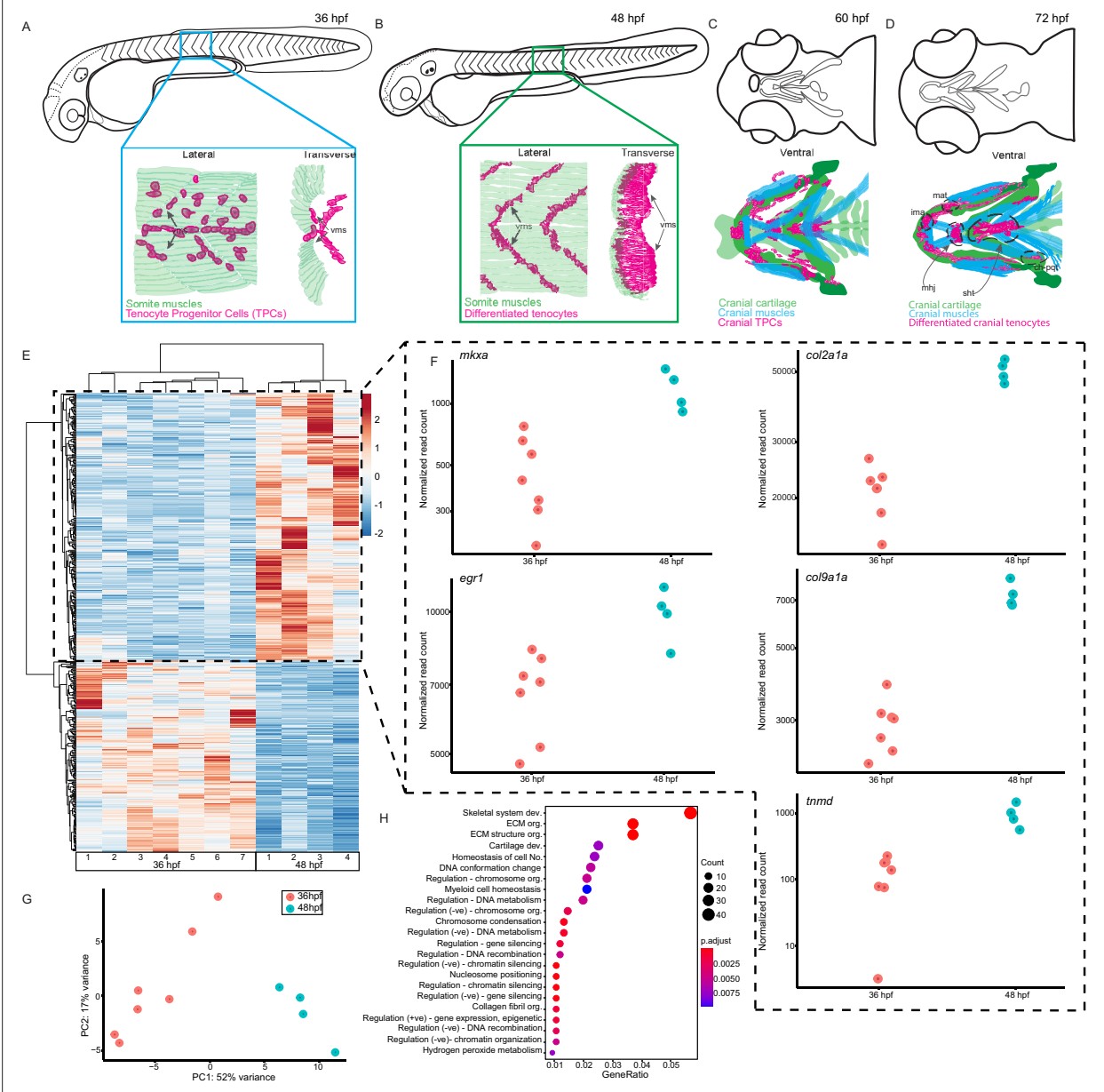

**Figure 1.** Onset of embryonic muscle contraction regulates transcription in tenocytes. (**A–D**) Diagrams depicting changes in tenocyte distribution and morphology during onset of trunk and cranial muscle contractions, (**A**) 36 hpf when twitching movements are sporadic and (**B**) 48 hpf when embryos become free swimming. Lateral views of 36 (**A**) and 48 hpf embryos (**B**). Insets show lateral and transverse views of migrating tenocyte progenitors (**A**) and differentiated tenocytes at somite boundaries with polarized, branched projections (**B**). Ventral views of the embryonic head in 60 hpf (**C**) and 72 hpf (**D**) embryos just prior to and during the onset of jaw movements. Cartilage (green), tenocytes (magenta), and muscles (cyan) showing tenocyte elongation, particularly in the sternohyoid tendon (sht) and condensation, as well as the mandibulohyoid junction (mhj). (**E**) Heatmaps from bulk RNA-sequencing (RNA-seq) showing the top 1000 differentially expressed genes (DEGs) between 36 and 48 hpf. p < 0.05. (**F**) Elevated expression of tenocyte marker genes *mkxa, tnmd,* and *egr1* and extracellular matrix (ECM) genes *col2a1a, col9a1a* in RNA-seq experiments at 48 hpf. Datapoints represent normalized read counts of single biological replicates at each color-coded timepoint (*n* = 7 for 36 hpf, *n* = 4 for 48 hpf). (**G**) Elevated expression of cartilage marker genes *col2a1a* and *col9a1a* in 48 hpf samples. (**H**) PCA of individual replicates showing separation of experimental conditions by timepoint. (**I**) GO analysis using Biological Process (BP) terms of top 2788 DEGs by adjusted p-value.

The online version of this article includes the following figure supplement(s) for figure 1:

**Figure supplement 1.** FACS gating thresholds for mCherry+ cells.

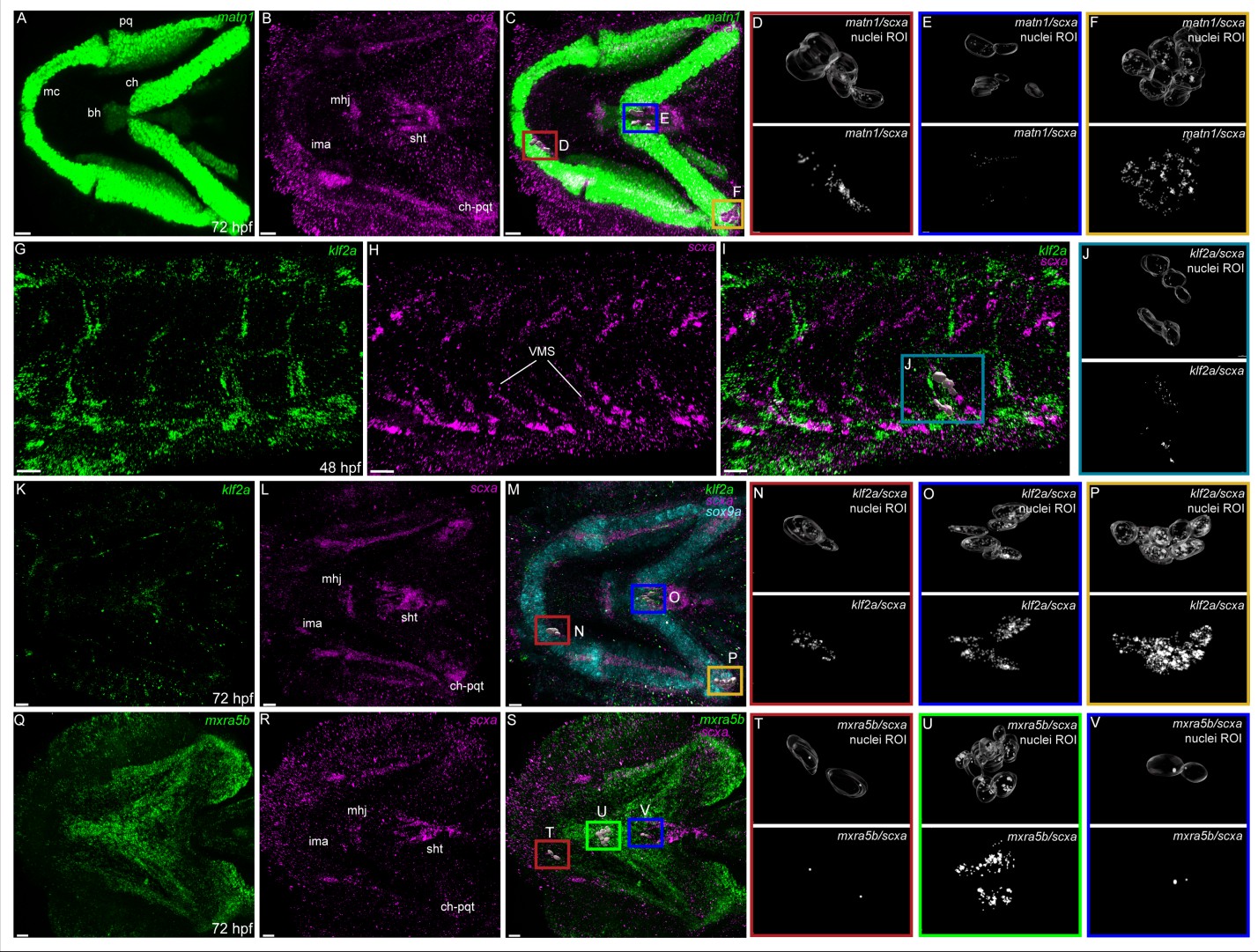

**Figure 2.** Expression of *matn1*, *klf2a*, and *mxra5b* with *scxa* in cranial and trunk tenocytes. Ventral cranial (**A–F, K–V**) and lateral trunk (**G–J**) views of 72 hpf (**A–F, K–V**) and 48 hpf (**G–J**) embryos showing *is*HCR of *matn1* (**A, C–F**), *klf2a* (**G, I–K, M–P**), and *mxra5b* (**Q, S–V**) in combination with *scxa* (**B–F, H–J, L–P, R–V**). (**D–F, J, N–P, T–V**) Higher magnification views of tenocyte nuclei in marked ROI. (**C, D, M, N, S, T**) ROI and panels outlined in magenta show magnified views of 3D volumes of tenocytes associated with imt. (**I, J**) ROI and panels outlined in cyan show magnified views of 3D volume of VMS tenocytes. (**C, E, M, O, S, V**) ROI and panels outlined in royal blue show magnified views of 3D volume of tenocytes associated with sht enthesis. (**C, F, M, P**) ROI and panels outlined in yellow show magnified views of 3D volumes of tenocytes associated with ch-pqt. (**S, U**) ROI and panels outlined in green show magnified views of 3D volumes of tenocytes associated with mhj. Each magnified view of ROI displays a translucent outline of the nuclear 3D volume with white puncta representing voxel colocalizations of *is*HCR as depicted by the colocalization function in Imaris (see Methods). mc – Meckel's cartilage, pq – palatoquadrate, ch – ceratohyal, bh – basihyal cartilage, ima – intermadibularis anterior tendon, mhj – mandibulohyoid junction, sht – sternohyoideus tendon, ch-pqt – ceratohyal-palatoquadrate tendon, sb – somite boundary. Scale bars = 20 µm.

The online version of this article includes the following figure supplement(s) for figure 2:

**Figure supplement 1.** *matn1*, *klf2a*, and *mxra5b* are expressed in musculoskeletal tissues of developing embryos.

**Figure supplement 2.** *matn1* is expressed in differentiating cranial tenocytes.

expression at 48 and 60 hpf in developing craniofacial and pectoral fin cartilages (*Figure 2—figure supplement 1A–C*). Differential expression of *matn1* in our tendon dataset could reflect expression in developing fibrocartilage enthesis progenitors closely associated with cartilages. To test this idea, we conducted fluorescent in situ Hybridization Chain Reaction (*is*HCR) for *scxa* and *matn1* at 51 hpf, slightly later than our RNA-seq samples to allow better visualization of differentiated chondrocytes, and 72 hpf after the onset of jaw movements. *scxa/matn1* co-expressing cells localized to the inter-mandibularis anterior tendon (ima) and sternohyoid tendon (sht), specifically in the entheses that

attach to meckels, anterior edge of the ceratohyal cartilages, and the posterior enthesis of the ceratohyal (ch-pqt), at 72 hpf (*Figure 2A–F*, *Figure 2—figure supplement 2A–D*).

For *klf2a*, chromogenic ISH revealed expression at VMS (somite boundaries) in the trunk at 48 hpf as well as developing pharyngeal arches and pectoral fins at 48 and 60 hpf (*Figure 2—figure supplement 1D–F*). This was confirmed by double *is*HCR of *klf2a* and *scxa* showing overlapping expression in tenocytes at VMS at 48 hpf (*Figure 2G–J*). *klf2a* expression was also detected in multiple cranial tendons at 72 hpf, most prominently in the entheses of the ima, sht, and ch-pqt (*Figure 2K–P*). This provides the first evidence for *klf2a* as an enthesis marker in craniofacial tendons, similar to Klf2 expression in developing mouse limb entheses (*Kult et al., 2021*; *Lu and Thomopoulos, 2013*; *Zelzer et al., 2014*).

*mxra5b* expression was first detected by chromogenic ISH at VMS near the horizontal myoseptum (HMS), which separates dorsal and ventral somites at 36 hpf, as well as in the notochord and cranial mesenchyme at 48 hpf (*Figure 2—figure supplement 1G, H*). Expression increased and extended along the VMS by 60 hpf (*Figure 2—figure supplement 1*). Double isHCR for *scxa* and *mxra5b*, detected *mxra5b* expression in cranial entheses (including ima, mhj, and sht – as well as others not shown), and in the mandibulohyoid junction tendon (mhj) in embryos at 72 hpf (*Figure 2Q–V*). Similar to *klf2a*, *mxra5b* expression has not been described in cranial connective tissues previously.

## Tenocyte-specific gene expression of *matn1*, *klf2a*, and *mxra5b* is regulated by muscle contraction

Since *matn1*, *klf2a*, and *mxra5b* were among the top DEGs in tenocytes at the onset of active swimming and persistent muscle activity, we reasoned that mechanical force regulates their expression. To test this, we performed Real Time Quantitative-PCR (RT-qPCR) in genetically paralyzed embryos. Relative expression of each gene was compared between wild-type (WT) embryos and homozygous mutants lacking the function of the voltage-dependent L-type calcium channel subtype beta-1 ($cacnb1^{-/-}$), which blocks muscle contraction (*Subramanian et al., 2018*; *Zhou et al., 2006*). At 48 hpf, all three genes were downregulated in $cacnb1^{-/-}$ mutants versus WT (*Figure 3—figure supplement 1A*). In contrast, at 72 hpf once jaw movements had begun, only *matn1* and *mxra5b* remained downregulated in $cacnb1^{-/-}$ embryos while *klf2a* expression increased (*Figure 3—figure supplement 1B*).

To confirm that loss of muscle contraction caused these transcriptional changes in tenocytes we injected *Tg(scxa:mCherry)* embryos at the 1-cell stage with full-length *alpha-bungarotoxin* mRNA (aBTX), which paralyzes embryos by irreversibly binding to acetylcholine receptors at neuromuscular synapses. Bulk RNA-seq of sorted *mCherry+* cells from whole aBTX-injected embryos at 48 hpf compared with WT uninjected controls (*Figure 3A*) identified 1450 DEGs. PC analysis clearly separated WT and aBTX biological replicates (*Figure 3B*, *Supplementary file 4*). 280 DEGs overlapped between both bulk RNA-seq runs (*Figure 3C*, *Supplementary file 5*). GO term analysis, using shinyGO (*Ge et al., 2020*), identified many of the same pathways downregulated in $cacnb1^{-/-}$ embryos, as well as others not previously implicated in tendon mechanotransduction. Seveal of these mapped to terms such as 'Focal Adhesion', including *rhoab*, *rock2a* (both part of Rho-ROCK signaling), and *col9a1a* (*Figure 3E*, *Supplementary file 6*) further implicating these as force dependent in tendons.

Comparisons of *matn1*, *klf2a*, and *mxra5b* expression between aBTX and WT versus our original 36 hpf versus free swimming 48 hpf RNA-seq experiment, revealed similar trends in expression. This suggests that the expression changes seen at embryonic stages (36 vs. 48 hpf) reflect tenocyte responsiveness to muscle contraction (*Figure 3D*). Further, comparing the 48 hpf WT versus $cacnb1^{-/-}$ mutant RT-qPCR with both bulk RNA-seq experiments, *matn1* and *mxra5b* expression were both consistently downregulated by paralysis, while *klf2a* expression was more variable across experiments (*Figure 3A, D*).

Having shown reproducible changes in their expression between bulk RNA-seq results, we next asked if variable recovery of muscle contractile forces differentially affects changes in *matn1*, *klf2a*, and *mxra5b* expression caused by paralysis. To test this, we used 90 ng/µl of aBTX , a concentration optimized to paralyze embryos only for the first 2 days of embryogenesis after which they gradually recover. Nearly all aBTX-injected embryos regained muscle contractions and were swimming at 48 hpf. We performed RT-qPCR on cDNA derived from these embryos and compared them to aBTX paralyzed (aBTX-P) and uninjected controls. We separated 48 hpf recovered embryos into two subgroups based on the extent of muscle contraction: (1) partially recovered (Twitching or aBTX-T), in which

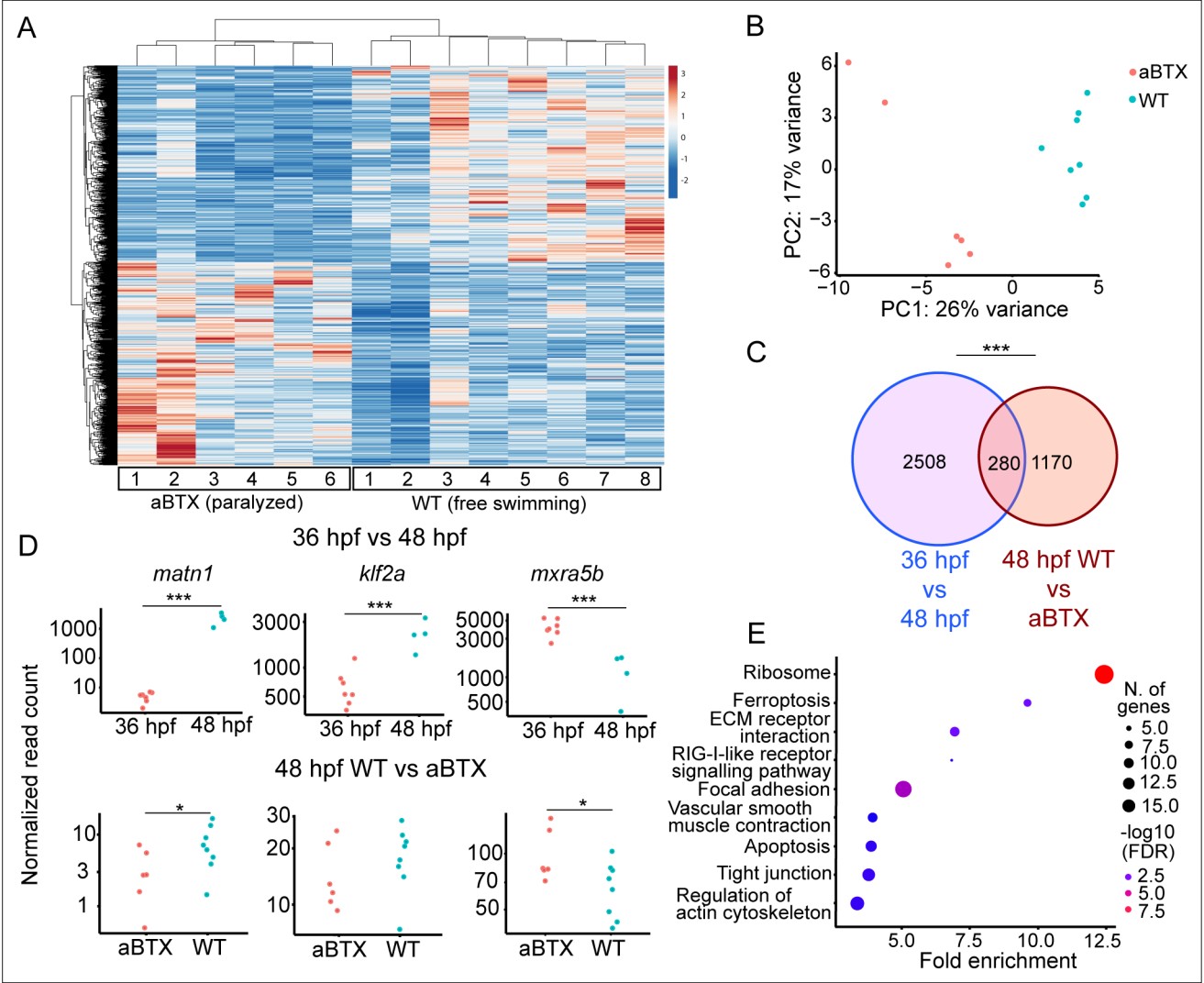

**Figure 3.** Paralysis regulates tenocyte gene expression in developing musculoskeletal system. (**A**) Heatmap of differentially expressed genes (DEGs) from bulk RNA-sequencing (RNA-seq) between WT and aBTX-injected (aBTX-inj) paralyzed 48 hpf embryos (force perturbed). (**B**) PCA of individual replicates WT versus aBTX-inj embryos' RNA-seq separate by experimental condition. (**C**) Venn diagram shows overlap of genes between developmental time-point and force perturbed RNA-seq experiments. (**D**) Comparison of normalized read counts between replicates of *matn1*, *klf2a*, and *mxra5b* in 36 versus 48 hpf and WT versus aBTX RNA-seq experiments. (**E**) KEGG pathway analysis plot shows enrichment of overlapping genes from (**C**). ns = not significant, *p < 0.05, ***p < 0.001.

The online version of this article includes the following figure supplement(s) for figure 3:

**Figure supplement 1.** Paralysis regulates gene expression of *matn1*, *klf2a*, and *mxra5b* in developing embryos.

embryos showed sporadic contractions of the trunk and pectoral fin muscles, similar to twitching 36 hpf embryos and (2) fully recovered (Recovered, or aBTX-R), in which embryos swam freely. At 48 hpf, RT-qPCR revealed significant global downregulation of *matn1* and *mxra5b* in αBTX paralyzed embryos compared to WT uninjected siblings, like *cacnb1*[−/−] mutant embryos (**Figure 3—figure supplement 1C**) and were upregulated in aBTX-T and aBTX-R embryos (**Figure 3—figure supplement 1D**). In contrast, *klf2a* was upregulated in paralyzed embryos, though this increase was also not statistically significant from WT controls (**Figure 3—figure supplement 1E**). These results, combined with those from RNA-seq, suggest that *matn1*, *klf2a*, and *mxra5b* transcription during development are regulated by muscle contraction.

To verify that these transcriptional changes occur specifically in tenocytes in response to force, we examined *matn1*, *klf2a*, and *mxra5b* expression in *scxa*-positive cells by *is*HCR with mCherry antibody staining of *Tg(scxa:mCherry)* fish using our αBTX paralysis-recovery experimental protocol (**Figure 4**,

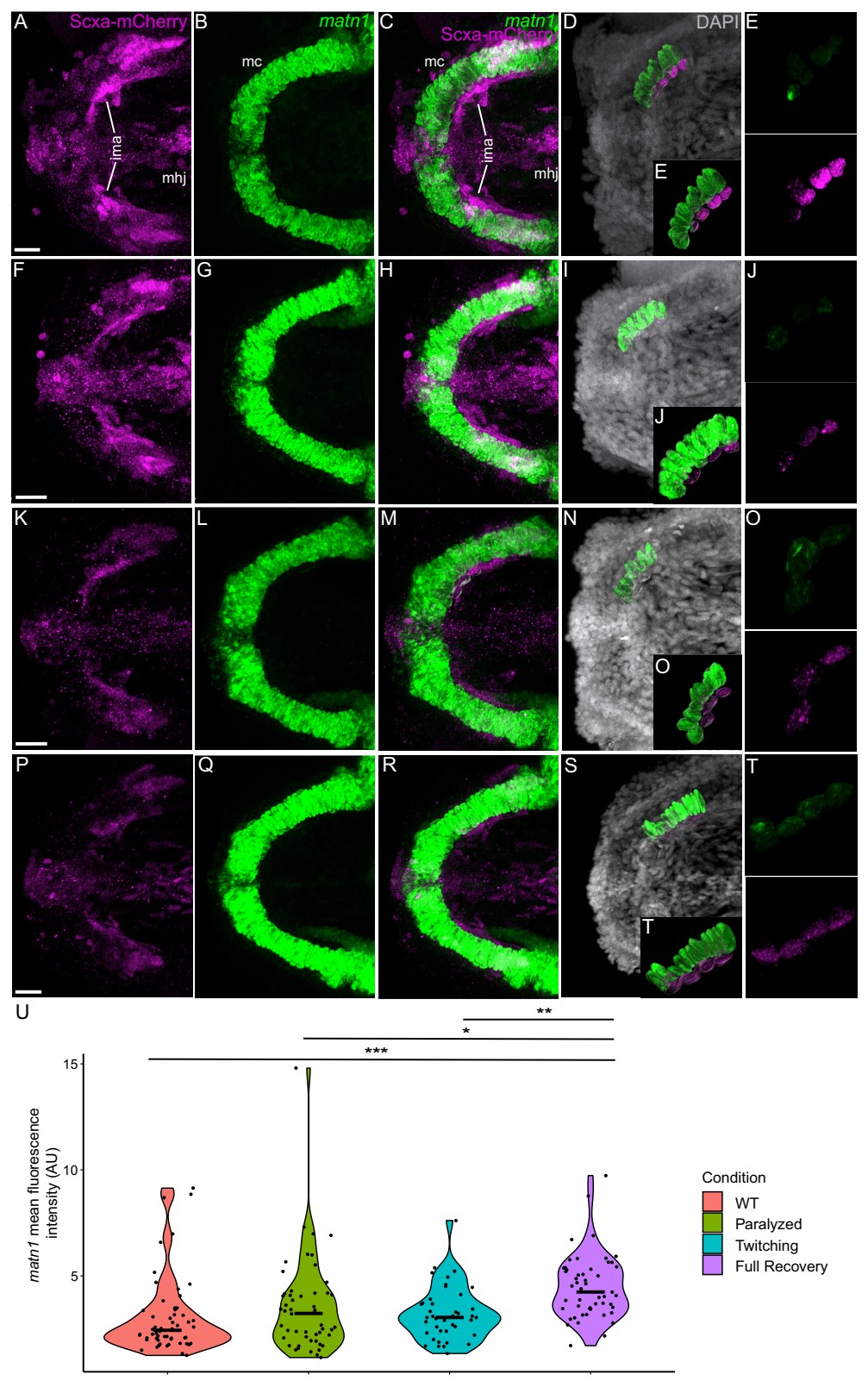

**Figure 4.** Mechanical force differentially regulates expression of *matn1* in ima enthesis tenocytes. Ventral views of Meckel's cartilage and associated tenocytes showing in situ Hybridization Chain Reaction (*is*HCR) of *matn1* (green) and anti-mCherry immunofluorescence (magenta) marking the tenocytes in *Tg(scxa:mCherry)* embryos at 72 hpf in WT uninjected (WT) (**A–E**), aBTX-inj (Paralyzed) (**F–J**), partially recovered aBTX-inj (Twitching) (**K–O**), and completely

*Figure 4 continued on next page*

*Figure 4 continued*

recovered aBTX-inj (Full Recovery) (**P–T**) conditions at ima enthesis. (**D, I, N, S**) Grayscale images showing nuclei stained with DAPI with ROIs showing isolated 3D volumes of chondrocytes (green) and enthesis tenocytes (magenta) based on DAPI signal. (**E, J, O, T**) Insets showing magnified views of the 3D volumes of tenocytes associated with ima enthesis depicting expression of *matn1* and stained for mCherry. (**U**) Violin plot showing changes in mean fluorescence intensity of *matn1* in ima enthesis tenocyte nuclei between WT (*n* = 8), Paralyzed (*n* = 8), Twitching (*n* = 6), and Full Recovery (*n* = 7) with ~8 nuclei measured per embryo. p-value calculated with linear mixed effects model with Tukey post hoc test. *p < 0.05, **p < 0.01, ***p < 0.001. Scale bars = 20 μm.

The online version of this article includes the following source data and figure supplement(s) for figure 4:

**Source data 1.** Measurements of *matn1 is*HCR signal intensity in ima enthesis tenocytes.

**Figure supplement 1.** Mechanical force regulates expression of *matn1* and *klf2a* in sht enthesis tenocytes.

**Figure supplement 1—source data 1.** Measurements of *matn1* and *klf2a is*HCR signal intensity in sht enthesis tenocytes.

---

*Figure 5*, *Figure 6*). Additionally, we quantified expression at multiple attachment regions across different tendons for each gene to determine if responses differed between spatially distinct tendons and by attachment type (e.g. enthesis vs. MTJ).

For *matn1*, we quantified expression by measuring its fluorescence intensity in individual tenocytes in 3D at the intermandibularis anterior (ima) enthesis where the ima attaches to meckel's (mc) cartilage and the sht enthesis at the anterior end of the ch cartilage (*Figure 4A–T*, *Figure 4—figure supplement 1A–L*; *Subramanian et al., 2023*). Cells were selected for quantification by their co-expression of *matn1* and Scxa and positions near chondrocytes expressing *matn1* alone and tenocytes expressing Scxa alone, as described previously (*Subramanian et al., 2023*). In these ima tenocytes, we found no significant difference in *matn1* expression between WT and paralyzed embryos, but increased expression in fully recovered (aBTX-R) embryos relative to WT, Paralyzed, and Twitching (aBTX-T) embryos (*Figure 4U*). Conversely, tenocytes of the sht enthesis showed no significant difference in expression across any of the conditions (*Figure 4—figure supplement 1M*).

We also examined fluorescence intensity in *scxa/mxra5b* or *scxa/klf2a* double positive tenocytes located at ima and sht entheses, as well as mhj and sht MTJs. *mxra5b* expression in the ima enthesis was significantly reduced in paralyzed, aBTX-T twitching, and remained low in aBTX-R fully recovered embryos compared to WT (*Figure 5—figure supplement 1Z*). However, in tenocytes of all other measured attachment sites (sht enthesis, mhj MTJ, sht MTJ), *mxra5b* expression returned to WT levels upon full recovery (*Figure 5*, *Figure 5—figure supplement 2M*, *Figure 5—figure supplement 3M*). *klf2a* expression in ima and sht entheses was significantly increased in paralyzed and aBTX-T embryos compared to WT and further increased upon full recovery (*Figure 4—figure supplement 1Z*, *Figure 5—figure supplement 1M*). However, unlike entheses, klf2a expression in sht MTJ tenocytes only increased significantly from twitching to full recovery, and in mhj MTJ tenocytes the pattern was much more variable, increasing upon paralysis, decreasing to WT levels at twitching, and re-increasing beyond WT levels at full recovery (*Figure 5—figure supplement 3*, *Figure 6U*).

To address functions of *matn1*, *klf2a*, and *mxra5b* in tenocytes we used multiplex CRISPR/Cas9 mutagenesis (*Wu et al., 2018*) to generate F0 CRISPants for *matn1*, *klf2a*, and *mxra5b*. While we did not observe obvious phenotypic defects in *matn1* and *klf2a* CRISPants, possibly due to genetic redundancy with other similar proteins, *Tg(scxa:mCherry)* embryos injected with four *mxra5b* gRNAs had qualitatively fewer trunk tenocytes when compared to uninjected controls (*Figure 7A*). Additionally, trunk VMS in *mxra5b* CRISPR-injected embryos displayed a wider sb angle (*Figure 7B*), although this may reflect a role for mxra5b in the notochord, where it is also expressed (*Figure 2—figure supplement 2G–I*). These results suggest that *mxra5b* may be required for embryonic axial tenocyte migration and/or differentiation.

## Discussion

Previous studies of mechanotransduction in tenocytes, particularly at the transcriptional level, have largely been limited to adult tendons or in vitro assays using mesenchymal stem cells. Few have addressed how functional differences in tendons are established during embryonic development. We report the first genome-wide survey of embryonic mechanoresponsive genes and transcriptional

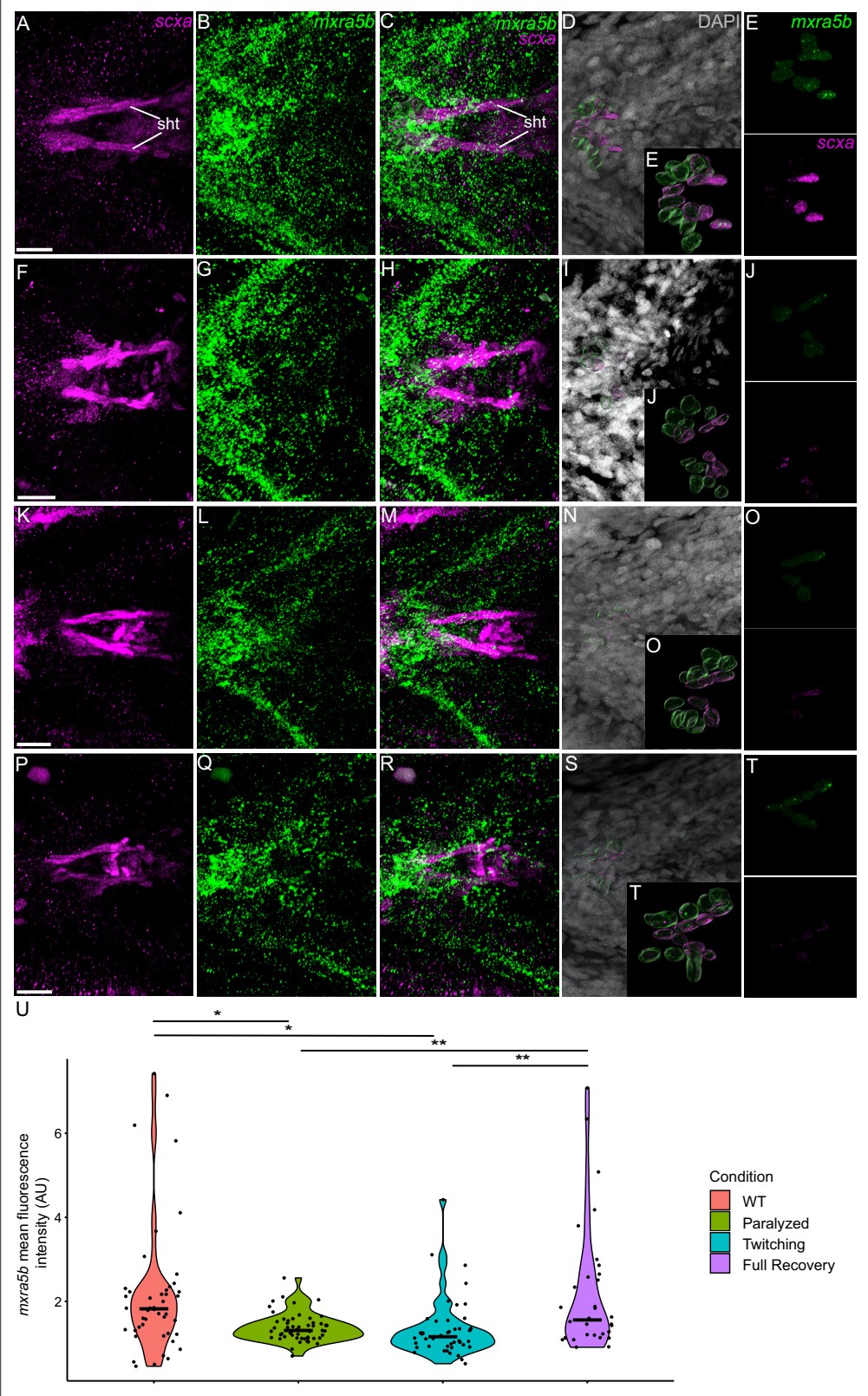

**Figure 5.** Mechanical force differentially regulates expression of *mxra5b* in sht enthesis tenocytes. Ventral views of ceratohyal (ch) cartilage and associated tenocytes showing in situ Hybridization Chain Reaction (*is*HCR) of *mxra5b* (green) and anti-mCherry immunofluorescence (magenta) marking the tenocytes in *Tg(scxa:mCherry)* embryos at 72 hpf in WT uninjected (WT) (**A–E**), aBTX-inj paralyzed (**F–J**), partially recovered aBTX-inj (Twitching)

*Figure 5 continued on next page*

*Figure 5 continued*

(**K–O**), and completely recovered aBTX-inj (Full Recovery) (**P–T**) conditions at sht enthesis. (**D, I, N, S**) Grayscale images showing nuclei stained with DAPI with ROIs showing isolated 3D volumes of chondrocytes (green) and sht enthesis tenocytes (magenta) based on DAPI signal. (**E, J, O, T**) Insets showing magnified views of the 3D volumes of tenocytes associated with sht enthesis depicting expression of *mxra5b* and stained for mCherry. (**U**) Violin plot showing changes in mean fluorescence intensity of *mxra5b* in sht enthesis tenocyte nuclei between WT (*n* = 7), Paralyzed (*n* = 8), Twitching (*n* = 8), and Full Recovery (*n* = 4) with ~8 nuclei measured per embryo. p-value calculated with linear mixed effects model with Tukey post hoc test. *p < 0.05, **p < 0.01. Scale bars = 20 μm.

The online version of this article includes the following source data and figure supplement(s) for figure 5:

**Source data 1.** Measurements of *mxra5b is*HCR signal intensity in sht enthesis tenocytes.

**Figure supplement 1.** Mechanical force differentially regulates expression of *mxra5b* and *klf2a* in ima enthesis tenocytes.

**Figure supplement 1—source data 1.** Measurements of *klf2a* and *mxra5b is*HCR signal intensity in ima enthesis tenocytes.

**Figure supplement 2.** Mechanical force regulates expression of *mxra5b* in mhj myotendinous junction tenocytes.

**Figure supplement 2—source data 1.** Measurements of *mxra5b is*HCR signal intensity in mhj MTJ tenocytes.

**Figure supplement 3.** Mechanical force differentially regulates expression of *mxra5b* and *klf2a* in sht myotendinous junction tenocytes.

**Figure supplement 3—source data 1.** Measurements of *mxra5b* and *klf2a is*HCR signal intensity in sht MTJ tenocytes.

---

responses across multiple tendon types. We identify three genes induced at the onset of muscle attraction and later maintained by contractile force (***Figure 8A***). Paralysis of zebrafish embryos alters expression of two ECM proteins in tenocytes, *matn1* and *mxra5b*, as well as the transcription factor *klf2a*. All three are expressed in cranial entheses, while *mxra5b* and *klf2a* are also expressed in trunk MTJs (***Figure 2***, ***Figure 8B***, ***Figure 2—figure supplement 1***, ***Figure 2—figure supplement 2***). Our previous studies have shown that in both tissues embryonic tenocytes in zebrafish acquire specialized morphologies and gene expression profiles as muscles first form functional attachments (***Subramanian et al., 2018***; ***Subramanian et al., 2023***; ***Subramanian and Schilling, 2015***). In contrast to classical studies of mature tendons these results suggest that cells with distinct enthesis or MTJ signatures arise in the embryo to fine-tune the ECM to match the functional demands of and forces exerted by individual muscles.

Classically tendon types and subdomains are distinguished by their collagen composition, and many collagens are direct Scx or Mkx transcriptional targets (***Bobzin et al., 2021***; ***Felsenthal and Zelzer, 2017***; ***Subramanian and Schilling, 2015***). This helps explain the gradient of stiffness and corresponding Scx/Sox9 expression within an enthesis (***Blitz et al., 2013***; ***Lu and Thomopoulos, 2013***; ***Subramanian et al., 2023***; ***Zelzer et al., 2014***). Our results highlight additional genes implicated in cartilage (i.e. *matn1*) and fibrocartilage (i.e. *KLF*) in entheseal tenocytes and their force responses. Though typically thought of as cartilage-specific, *matn1* and its relatives have been reported in single-cell RNA-seq (scRNA-seq) analyses of adult tenocytes and fibrocartilage (***Kaji et al., 2021***). We find that zebrafish *matn1* regulation differs between entheses that form at different stages (***Figure 4***, ***Figure 4—figure supplement 1***). Whereas paralyzed embryos at both twitching and swimming stages show reduced tenocyte *matn1* expression (***Figure 3D***), our *is*HCR data reveal that expression only rebounds after full recovery of muscle contraction in the ima enthesis (***Figure 8B***, ***Figure 4—figure supplement 1***; ***Figure 4***). These spatial and temporal differences support our hypothesis that these are bona fide embryonic entheseal tenocytes specified at the edges of cartilages as muscles first attach (***Subramanian et al., 2023***). They are also consistent with studies showing that *matn1* transcription is upregulated upon mechanical load in cultured chondrocytes (***Chen et al., 2016***). Chondrocyte ECM becomes disorganized in $Matn1^{-/-}$ mutant mice exposed to mechanical loads after medial meniscus destabilization surgery (***Chen et al., 2016***; ***Li et al., 2020***). Our data implicate *matn1* in tendon/fibrocartilage mechanotransduction and in the initial establishment of ECM stiffness gradients at entheses during embryogenesis (***Figure 4***, ***Figure 4—figure supplement 1***; ***Lu and Thomopoulos, 2013***).

*Mxra5* (also known as *adlican*) encodes a secreted proteoglycan implicated in cell–cell adhesion and ECM remodeling, mainly in the context of colorectal and other cancers (***He et al., 2015***; ***Wang***

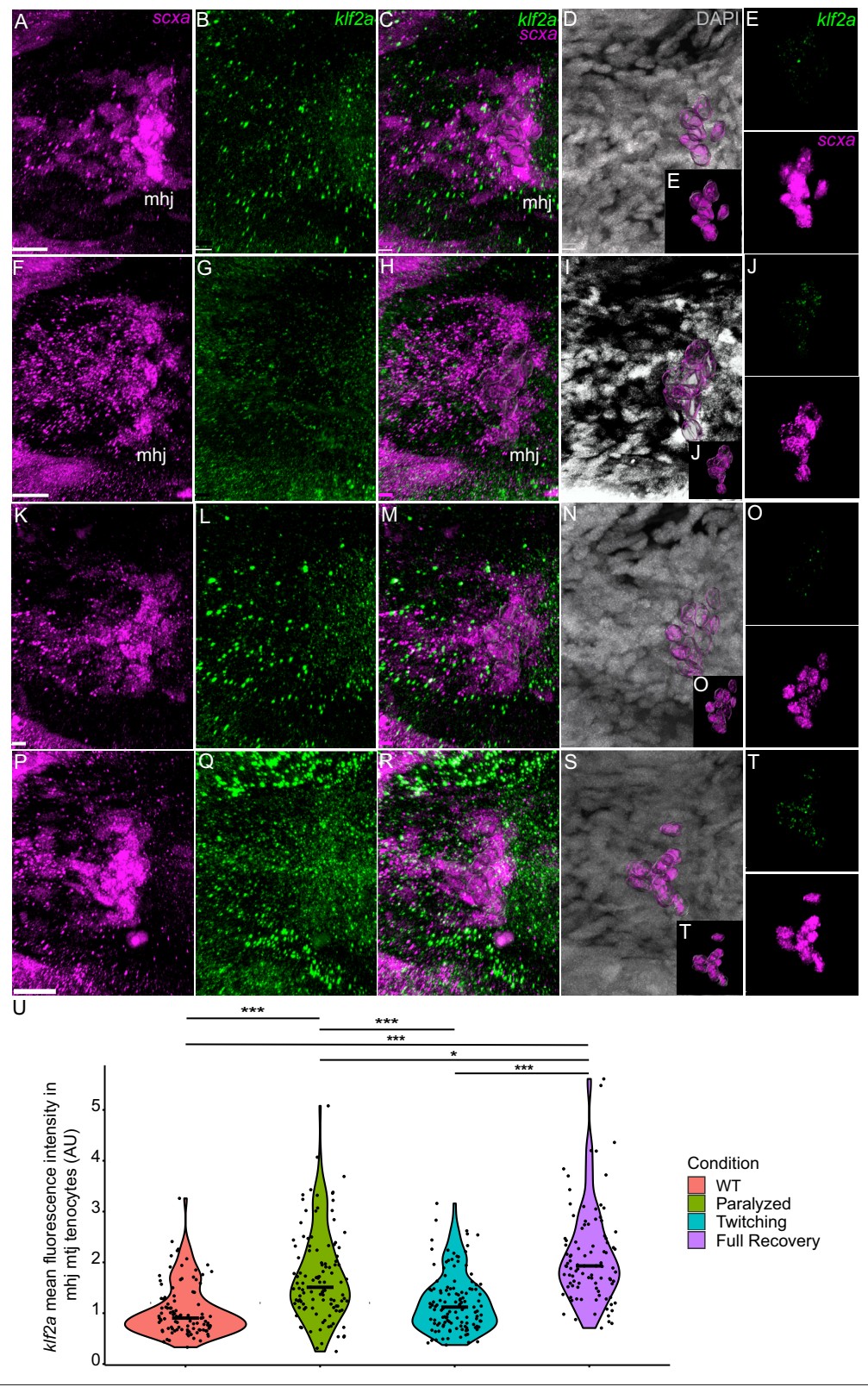

**Figure 6.** Mechanical force regulates expression of *klf2a* in mhj myotendinous junction tenocytes. Ventral views of mandibulohyoid junction (mhj), myotendinous junction (MTJ) associated tenocytes showing in situ Hybridization Chain Reaction (*is*HCR) of *klf2a* (green) and anti-mCherry immunofluorescence (magenta) marking the tenocytes in *Tg(scxa:mCherry)* embryos at 72 hpf in WT uninjected (WT) (**A–E**), aBTX-inj (Paralyzed) (**F–J**), partially recovered

*Figure 6 continued on next page*

*Figure 6 continued*

aBTX-inj (Twitching) (**K–O**), and completely recovered aBTX-inj (Full Recovery) (**P–T**) conditions. (**D, I, N, S**) Grayscale images showing nuclei stained with DAPI with ROIs showing isolated 3D volumes of mhj tenocytes (magenta) based on DAPI signal. (**E, J, O, T**) Insets showing magnified views of the 3D volumes of tenocytes associated with mhj MTJ depicting expression of *klf2a* and stained for mCherry. (**U**) Violin plot showing changes in mean fluorescence intensity of *klf2a* in mhj MTJ tenocyte nuclei between WT (*n* = 17), Paralyzed (*n* = 15), Twitching (*n* = 14), and Full Recovery (*n* = 11) with ~10 nuclei measured per embryo. p-value calculated with linear mixed effects model with Tukey post hoc test. *$p < 0.05$, ***$p < 0.001$. Scale bars = 20 μm.

The online version of this article includes the following source data for figure 6:

**Source data 1.** Measurements of *klf2a* isHCR signal intensity in mhj MTJ tenocytes.

---

*et al., 2013*). *Mxra5* is expressed in tendons and other connective tissues of developing chick embryos as well as human fibroblasts (*Chondrogianni et al., 2004*; *Robins and Capehart, 2018*). We find that zebrafish *mxra5b* expression is downregulated in all tenocytes at the onset of embryonic muscle contraction, unlike *matn1* (*Figures 3 and 5*, *Figure 5—figure supplements 1–3*). Consistent with a force-responsive gene, MXRA5 is inhibited by TGF-β1 (*Poveda et al., 2017*), and associated with migration of dental pulp stem cells (*Yoshida et al., 2023*). Our results provide the first evidence for regulation of *mxra5b* transcription in tenocytes by mechanotransduction. However, despite reductions in *mxra5b* levels overall with loss of active muscle contraction, our *is*HCR results suggest that these changes differ between distinct tendons and force conditions (*Figure 5*). For example, in the ima enthesis, paralysis downregulates *mxra5b* expression, with little rebound after recovery (*Figure 8B*; *Figure 5—figure supplement 1*). In contrast, at other entheses and MTJs *mxra5b* expression returns to WT levels upon full recovery after paralysis (*Figures 5 and 8*, *Figure 5—figure supplements 2 and 3*). *mxra5b* expression may require continuous mechanical activation, levels of which differ between tendons as well as entheses or MTJs (*Figure 8B*). This heterogeneity may help explain differences between our RNA-seq results for *mxra5b* and *is*HCR expression data, since the RNA-seq experiments were performed on FAC-sorted tenocytes of all tendons (*Figure 3*).

Similar to *matn1* and *mxra5b*, (1) zebrafish *klf2a* expression localizes to embryonic cranial entheses, (2) its transcription increases in tenocytes at the onset of muscle contraction, and (3) these responses vary between spatially distinct tendons and tendon subdomains (*Figure 8B*, *Figure 3*, *Figure 4—figure supplement 1*, *Figure 5—figure supplements 1 and 3*, *Figure 6*). Mammalian Klf2 and Klf4

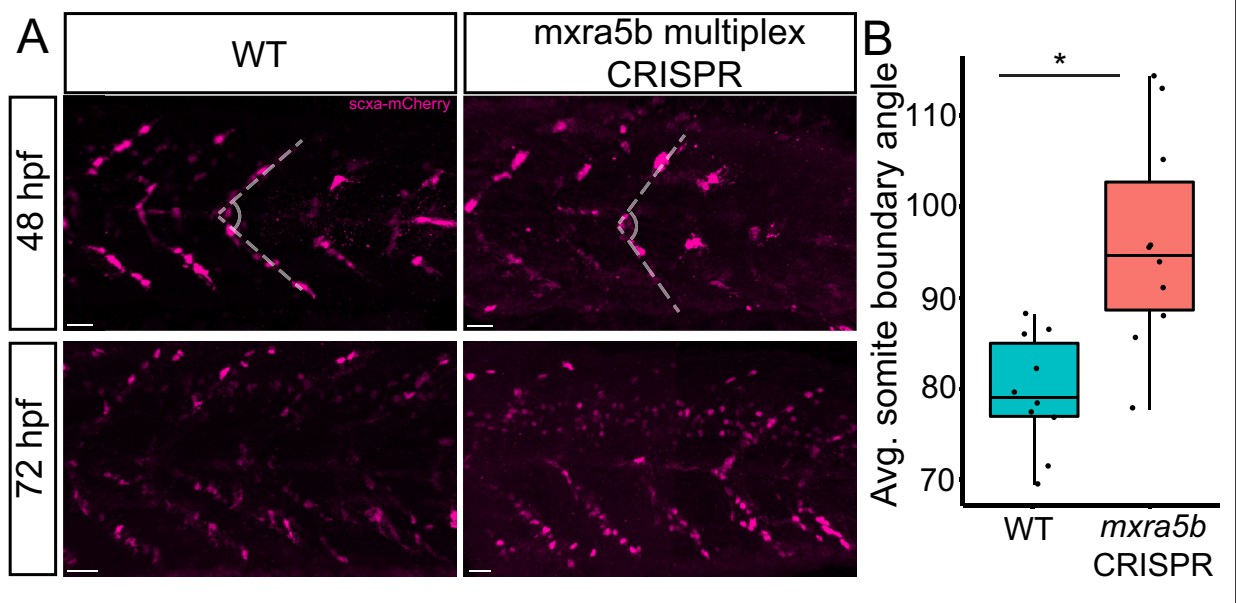

**Figure 7.** Loss of mxra5b function affects somite boundary structure. (**A**) Lateral views of WT and *mxra5b* multiplex CRISPants at 48 and 72 hpf Tg(*scx:mCherry*) embryos stained with anti-mCherry to show tenocytes at the somite boundary (SB). (**B**) Quantification of somite boundary angle measurements of 48 hpf WT or *mxra5b* multiplex CRISPant embryos. p-value calculated with Watson's U2 test. *$p < 0.05$.

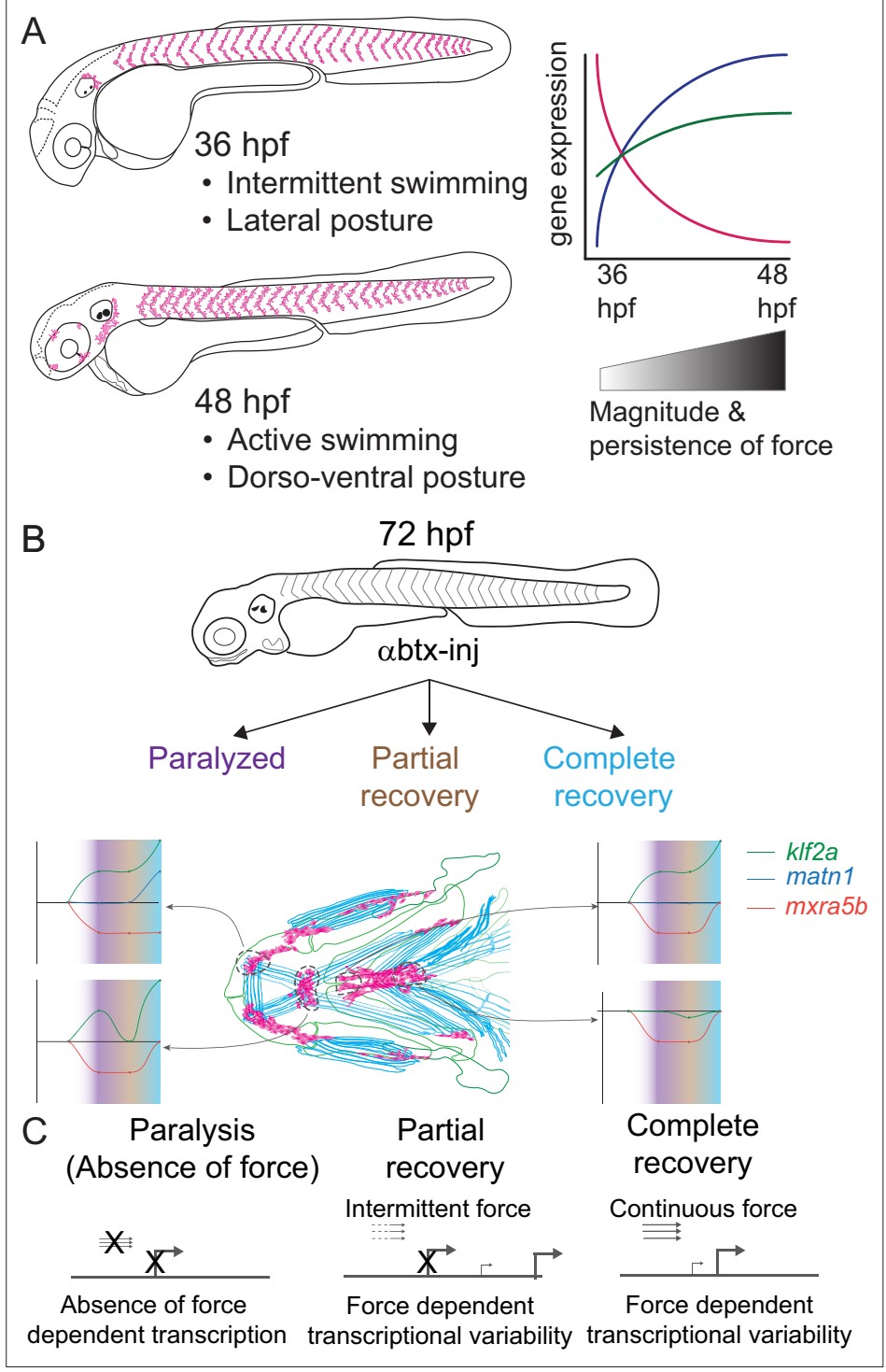

**Figure 8.** Model depicting role of mechanical force in regulating expression of genes in tenocytes during onset of active muscle contraction. (**A**) Cartoon showing role of force in regulating tenocyte morphogenesis and gene expression in tenocytes between 36 and 48 hpf stages correlating with onset of active swimming The variability in gene expression is related to increase in both magnitude and persistence of muscle contraction force. (**B**) Representative model summarizing the multifaceted role of muscle contractile force on expression dynamics of *matn1*, *klf2a*, and *mxra5b* genes in cranial tendon attachments. (**C**) Force-responsive gene expression is more nuanced than a binary on/off control.

have been implicated in cell differentiation at tendon-bone entheses (*Kult et al., 2021*). Cranial teno-cytes in zebrafish upregulate *klf2a* upon recovery from paralysis (*Figures 3 and 6*, *Figure 4—figure supplement 1*, *Figure 5—figure supplements 1 and 3*), though there are discrepancies between *is*HCR, bulk RNA-seq, and RT-qPCR measurements. These may reflect the fact that *klf2a* is also expressed in other tissues, such as embryonic vascular and endocardial cells (*Figure 2*; *Goddard et al., 2017*) or differences in expression between trunk and cranial tenocyte populations. The *is*HCR data show distinct entheseal *klf2a* and MTJ expression patterns (*Figure 4—figure supplement 1*, *Figure 5—figure supplements 1 and 3*, *Figure 6* and *Figure 8B*). Klf2-binding sites have been iden-tified upstream of ECM genes such as *Col5* in sorted entheseal tenocytes (*Kult et al., 2021*). *Klf2* expression is also upregulated by fluid forces in endocardial cells leading to fibronectin synthesis (*Boselli et al., 2015*; *Lee et al., 2006*; *Steed et al., 2016*). Thus**,** force-dependent *klf2a* expression may be critical for tissue-specific ECM remodeling in many contexts.

Together, our bulk RNA-seq analysis of embryonic zebrafish tenocytes and their transcriptional responses to muscle contraction: (1) identifies new regulators of tenocyte–ECM, going beyond the better studied collagens, and (2) highlights the importance of considering developmental events that specify the mechanical properties of tendons as they form. Genes such as *matn1*, *mxra5b*, and *klf2a* show unique expression profiles and changes due to perturbation of muscle contraction, both during normal embryonic development and in response to paralysis (*Figure 8B*). The presence of these genes in embryonic tendons and their responses to force during normal development versus recovery from paralysis raises questions as to whether the mechanisms that initially establish these structures differ from those that control their maintenance (*Figure 8B*). Though cell–ECM feedback mechanisms have been studied in controlled 3D microenvironments in vitro, extrapolating these mechanisms into an understanding of in vivo biological processes like development and tissue homeostasis is neces-sary (*Saraswathibhatla et al., 2023*). Given the large variation of cell–ECM feedback mechanisms throughout embryonic development, understanding specific tenocyte–ECM interactions will require novel approaches to measuring the effect of varying (1) ECM microenvironment protein compositions, or local 'matrisomes', on tenocyte gene expression and (2) intrinsic gene expression patterns of hetero-geneous tenocyte populations spatially and functionally. Single-cell approaches (e.g. scRNA-seq) at different developmental stages and in the presence or absence of force, will provide a clearer under-standing of how individual spatially and functionally distinct tenocyte populations respond to force in development. Integrating such knowledge of the basic biology of tenocytes at multiple scales will be essential for developing a better picture of tenocyte–ECM interactions at individual tendons, paving the path to advance personalized translational therapies for tendon injuries.

# Materials and methods

## Key resources table

| Reagent type (species) or resource | Designation | Source or reference | Identifiers | Additional information |
|---|---|---|---|---|
| Strain (*Danio rerio*) | AB | Schilling lab | RRID:NCBITaxon_7955 | |
| Genetic reagent (*Danio rerio*) | *Tg(scxa:mCherry)* | Galloway lab | fb301Tg; RRID:ZFIN_ZDB-GENO-180925-6 | *scx* BAC transgenic in AB background |
| Genetic reagent (*Danio rerio*) | *cacnb1⁻/⁻; Tg(scxa:mCherry)* | Schilling lab | lr1092;fb301; RRID:ZFIN_ZDB-ALT-191023-1 | *cacnb1* mutant in *Tg(scx:mCherry)* background |
| Sequence-based reagent | T7 sequence-tagged primers | This paper | *Supplementary file 7* | 2 mM final concentration |
| Commercial assay or kit | Protoscript II first strand cDNA synthesis kit | New England Biolabs | Cat # E6560 | |
| Commercial assay or kit | T7 RNA polymerase | Millipore Sigma (Roche) | Cat # 10881767001 | |
| Commercial assay or kit | Monarch Total RNA Miniprep kit | New England Biolabs | Cat # T2010S | |

*Continued on next page*

*Continued*

| Reagent type (species) or resource | Designation | Source or reference | Identifiers | Additional information |
|---|---|---|---|---|
| Commercial assay or kit | DIG RNA labeling mix | Millipore Sigma (Roche) | Cat # 11277073910 | |
| Commercial assay or kit | MEGAshortscript T7 transcription kit | Thermo Fisher Scientific (Invitrogen) | Cat # AM1354 | |
| Commercial assay or kit | Luna Universal qPCR master mix | New England Biolabs | Cat # M3003S | |
| Commercial assay or kit | Zirconium beads | Benchmark Scientific | Cat # D1032-10 | |
| Commercial assay or kit | RNEasy Micro Kit | QIAGEN | Cat # 74004 | |
| Commercial assay or kit | 40 µm filter | Pluriselect-USA | Cat # 43-10040-50 | |
| Commercial assay or kit | HCR Buffers (v3.0) | Molecular Instruments | | Hybridization buffer, Wash buffer, Amplifier buffer |
| Antibody | Anti-Digoxigenin-AP, Fab fragments | Millipore Sigma (Roche) | Cat # 11093274910 RRID:AB_514497 | 1:2000 |
| Antibody | Rat monoclonal anti-mCherry antibody | Invitrogen (Thermo Fisher Scientific) | Cat # M11217 RRID:AB_2536611 | 1:500 |
| Antibody | Chicken polyclonal anti-GFP antibody | abcam | Cat # ab13970 RRID:AB_300798 | 1:1000 |
| Antibody | Mouse monoclonal anti-myosin heavy chain antibody | Developmental Studies Hybridoma Bank (DHSB) | Cat # A4.1025 RRID:AB_528356 | 1:200 |
| Antibody | Alexa Fluor 594 AffiniPure F(ab')$_2$ Fragment Donkey polyclonal anti-Rat IgG (H+L) | Jackson ImmunoResearch Laboratories | Cat # 712-586-153 RRID:AB_2340691 | 1:1000 |
| Antibody | Alexa Fluor 488 AffiniPure F(ab')$_2$ Fragment Donkey polyclonal anti-Chicken IgY IgG (H+L) | Jackson ImmunoResearch Laboratories | Cat # 703-546-155 RRID:AB_2340376 | 1:1000 |
| Antibody | Alexa Fluor 647 AffiniPure F(ab')$_2$ Fragment Donkey polyclonal anti-Mouse IgG (H+L) | Jackson ImmunoResearch Laboratories | Cat # 715-606-151 RRID:AB_2340866 | 1:1000 |
| Chemical compound, drug | Nitro Blue Tetrazolium chloride solution (NBT) | Millipore Sigma (Roche) | Cat # 11383213001 PubChem CID: 9281 | |
| Chemical compound, drug | 5-Bromo-4-chloro-3-indolyl phosphate solution | Millipore Sigma (Roche) | Cat # 11383221001 PubChem CID: 81059 | |
| Chemical compound, drug | Ethylenediaminetetraacetic acid disodium salt | Millipore Sigma (Roche) | Cat # E5134 PubChem CID: 8759 | |
| Chemical compound, drug | Calcium chloride hexahydrate | Millipore Sigma (Roche) | Cat # 21108 PubChem CID: 6093252 | |
| Chemical compound, drug | Dulbecco's phosphate-buffered saline (DPBS) 1× | Thermo Fisher Scientific (Gibco) | Cat # 14190144 | |
| Chemical compound, drug | Agarose low gelling temperature | Millipore Sigma (Sigma-Aldrich) | Cat # A9414 | |
| Chemical compound, drug | SSC buffer 20× | Millipore Sigma (Sigma-Aldrich) | Cat # S6639-1L | |
| Chemical compound, drug | DAPI | Millipore Sigma (Sigma-Aldrich) | Cat # D9542 PubChem CID: 2954 | |

*Continued on next page*

*Continued*

| Reagent type (species) or resource | Designation | Source or reference | Identifiers | Additional information |
|---|---|---|---|---|
| Sequence-based reagent | matn1-B1 | Molecular instruments | NM_001099740.2 | 20 probe set |
| Sequence-based reagent | mxra5b-B1 | Molecular instruments | XM_017357865.2 | 20 probe set |
| Sequence-based reagent | mxra5b-B1 | Molecular instruments | XM_017357865.2 | 20 probe set |
| Sequence-based reagent | klf2a-B3 | Molecular instruments | NM_131856.3 | 20 probe set |
| Sequence-based reagent | scxa-B2 | Molecular instruments | NM_001083069 | 18 probe set |
| Sequence-based reagent | B1-h1&h2- Alexa Fluor 488 amplifier hairpins | Molecular instruments | | HCR RNA-FISH (v3.0) |
| Sequence-based reagent | B2-h1&h2- Alexa Fluor 546 amplifier hairpins | Molecular instruments | | HCR RNA-FISH (v3.0) |
| Sequence-based reagent | B3-h1&h2- Alexa Fluor 647 amplifier hairpins | Molecular instruments | | HCR RNA-FISH (v3.0) |
| Other | BD FACSAria II Cell Sorter | Becton, Dickinson and Company | RRID:SCR_018934 | |
| Other | Bioanalyzer 2100 instrument | Agilent | RRID:SCR_018043 | |
| Other | HiSeq 4000 sequencing system | Illumina | RRID:SCR_016386 | |
| Other | NextSeq 550 system | Illumina | RRID:SCR_016381 | |
| Other | LightCycler 480 Real Time PCR System | Roche | RRID:SCR_018626 | |
| Other | SP8 Lightning Confocal microscope | Leica | RRID:SCR_018169 | |
| Other | Zeiss Axioplan 2 imaging system | Zeiss | RRID:SCR_020918 | |
| Other | MicroPublisher color RTV-5.0 CCD camera | QImaging | | |
| Other | BeadBug 3 microtube homogenizer | Benchmark Scientific | Cat # D1030 | |
| Peptide, recombinant protein | Protease (Subtilisin Carlsberg) from *Bacillus licheniformis* | Millipore Sigma (Sigma-Aldrich) | Cat # P5380 UniProtKB: P00780.SUBC_BACLI | |
| Peptide, recombinant protein | Collagenase Type IV from *Hathewaya histolytica* (*Clostridium histolyticum*) | Thermo Fisher Scientific (Gibco Life technologies) | Cat # 17104019 | |
| Peptide, recombinant protein | Deoxyribonuclease I (DNase I) from bovine pancreas | Millipore Sigma (Roche) | Cat # 10104159001 UniProtKB: P00639.DNAS1_BOVIN | |
| Peptide, recombinant protein | Bovine serum albumin stock solution (10%) | Miltenyi Biotec | Cat # 130-091-376 | |
| Recombinant DNA reagent | pmtb-t7-alpha-bungarotoxin | Addgene (Megason lab) | Cat # 69542 RRID:Addgene_69542 | |
| Software, algorithm | Spliced Transcripts Alignment to a Reference (STAR) v2.5.2a | Dobin lab | RRID:SCR_004463 | |
| Software, algorithm | Smart-seq2 single sample pipeline | Broad Institute | RRID:SCR_021228 | |
| Software, algorithm | RSEM v1.2.31 | Dewey lab | RRID:SCR_000262 | |
| Software, algorithm | DESeq2 v1.30.1 | Anders lab | RRID:SCR_015687 | |
| Software, algorithm | ClustVis | Vilo lab | RRID:SCR_017133 | |
| Software, algorithm | ClusterProfiler R package | Qing-Yu lab | RRID:SCR_016884 | |
| Software, algorithm | ShinyGO | Ge lab | RRID:SCR_019213 | |

*Continued on next page*

*Continued*

| Reagent type (species) or resource | Designation | Source or reference | Identifiers | Additional information |
|---|---|---|---|---|
| Software, algorithm | VennDiagram v1.7.3 | Boutros lab | RRID:SCR_002414 | |
| Software, algorithm | GeneOverlap v1.26.0 | Shen lab | RRID:SCR_018419 | |
| Software, algorithm | LightCycler Software | Roche | RRID:SCR_012155 | |
| Software, algorithm | Zeiss Zen Microscopy software | Zeiss | RRID:SCR_013672 | |
| Software, algorithm | Leica Application Suite X | Leica | RRID:SCR_013673 | |
| Software, algorithm | Imaris | Bitplane | RRID:SCR_007370 | |
| Other | Optical Biology Core at UCI | Department of Developmental Biology, UCI | RRID:SCR_026614 | Core facility |
| Other | Genomics Research and Technology Hub Core at UCI | Department of Biological Chemistry, UCI | RRID:SCR_026615 | Core facility |
| Other | Flow Cytometry Core at UCI | Stem Cell Research Center, UCI | RRID:SCR_026616 | Core facility |

## Zebrafish embryos, transgenics, and mutants

WT zebrafish (AB strain; RRID:NCBITaxon_7955), *TgBAC(scxa:mCherry)$^{fb301}$* transgenics referred to as *Tg(scxa:mCherry)* (RRID:ZFIN_ZDB-GENO-180925-6), or *cacnb1$^{ir1092/ir109}$;fb301Tg* (referred to as *cacnb1$^{-/-}$* mutants; RRID:ZFIN_ZDB-ALT-191023-1) embryos were raised in embryo medium at 28.5°C (*Westerfield, 2000*) and staged as described (*Kimmel et al., 1995*). Craniofacial musculoskeletal structures were identified and annotated as described previously (*Schilling and Kimmel, 1997*; *Subramanian et al., 2023*). All protocols performed on embryos and adult zebrafish in this study had prior approval from the IACUC at UC Irvine (protocol # AUP-23-099).

## In situ hybridization

Digoxigenin-labeled antisense RNA probes for *matn1*, *klf2a*, and *mxra5b* were generated using T7 sequence-tagged primers (*Supplementary file 7*). Total embryo RNA was extracted from 72 hpf WT embryos using Trizol (Invitrogen 15596026) and a Monarch Total RNA Miniprep kit (New England Biolabs (NEB) T2010S. cDNA was synthesized using oligo dT primers and a ProtoScript II First Strand cDNA Synthesis Kit (NEB E6560) and used as a template to synthesize RNA probes using T7 RNA polymerase (Roche, 10881767001) and DIG RNA labeling mix (Roche, 11277073910). Whole-mount ISH was performed with anti-DIG-AP fragments (Roche, 11093274910) at 1:2000 dilution, as described in *Thisse and Thisse, 2008*.

## In situ hybridization chain reaction (*is*HCR) and immunohistochemistry

*is*HCR probes were designed by Molecular Instruments Inc (Los Angeles, CA) and whole mount *is*HCR was performed with amplifiers/probes obtained from Molecular Instruments according to the *is*HCR v3.0 protocol as described (*Choi et al., 2014*; *Subramanian et al., 2023*; *Trivedi et al., 2018*). Probes/amplifier combinations used were: *matn1* (NCBI ref # NM_001099740.2); *mxra5b* (NCBI ref # XM_017357865.2) in B1 with B1 Alexa Fluor 488, *scxa* (NCBI ref # NM_001083069) in B2 with B2 Alexa Fluor 546, *klf2a* (NCBI ref # NM_131856.3) in B3 with B3 Alexa Fluor 647.

Whole embryo immunohistochemistry was performed as described in *Subramanian et al., 2018*. Primary antibodies used: rat monoclonal anti-mCherry (Molecular Probes – 1:500 dilution, M11217, RRID:AB_2536611), chicken anti-GFP (Abcam – 1:1000 dilution, ab13970, RRID:AB_300798), mouse anti-myosin heavy chain (Developmental Hybridoma – 1:250, A1025, RRID:AB_528356). Secondary antibodies used: Alexa Fluor 594 conjugated donkey anti-rat IgG (Jackson ImmunoResearch – 1:1000 dilution, 712-586-153, RRID:AB_2340691), Alexa Fluor 488 conjugated donkey anti-chicken IgY (Jackson Immunoresearch, 1:1000 dilution, 703-546-155, RRID:AB_2340376), Alexa Fluor 647 conjugated donkey anti-mouse IgG (Jackson Immunoresearch, 1:1000 dilution, 715-606-151, RRID:AB_2340866).

## Embryo dissociation and FAC sorting

For WT 36–48 hpf bulk RNA-sequencing (bulk RNA-seq), transgenic *Tg(scxa:mCherry)* zebrafish embryos were dissociated using collagenase IV (Gibco, 17104019) at a concentration of 6.25 mg/ml without trypsin addition at a temperature of 28°C for roughly 40 min, homogenizing every 5 min using a P1000 pipette as described in *Barske et al., 2016*. Cells were then filtered through a 40-µm filter (Pluriselect-usa, 43-10040-50). Dissociated cell suspensions were sorted on a BD FACS Aria II cell sorter (RRID:SCR_018934) at the Flow Cytometry Core facility (RRID:SCR_026616). mCherry-positive cells were gated and sorted for those expressing at high levels.

For aBTX-injected 48 hpf bulk RNA-seq, transgenic *Tg(scxa:mCherry)* embryos, aBTX- or uninjected siblings, were dissociated using Subtilisin A cold-active protease in a stock solution consisting of: 5 µl of 1 M CaCl$_2$ (Sigma 21108; PubChem CID: 6093252), 100 µl of protease stock solution (100 mg of Bacillus licheniformis protease (Sigma P5380; UniProtKB: P00780.SUBC_BACLI) solubilized in 1 ml of Ca and Mg free PBS), 889 µl of 1× DPBS (Thermo Fisher 14190144), 1 µl of 0.5 M EDTA (Sigma E5134; PubChem CID: 8759), and 5 µl of DNAse I (Roche 10104159001; UniProtKB: P00639.DNAS1_BOVIN) stock (25 U/µl in PBS, stored at –80°C) adapted from *O'Flanagan et al., 2019*. Embryos were triturated once every 2 min for 15 s using a wide bore 1 ml pipette. Every 15 min, the tissue suspension was checked under a dissecting scope to verify dissociation. Full dissociation took ~30 min per sample, and samples were subsequently run through a 40-µm filter to separate dissociated cells from clumps of aggregated undissociated tissue/ECM and washed with 10 ml of PBS/BSA (0.01% BSA in PBS, made fresh on the day of dissociation) and transferred to a 15 ml conical tube. Cells were centrifuged at 600 × *g* for 5 min at 4°C, supernatant discarded, and cells were resuspended in 1 ml of ice-cold PBS/BSA before being placed on ice (*Subramanian et al., 2025*). Cells expressing high levels of mCherry+ cells were gated and sorted on a BD FACS Aria II cell sorter.

## Bulk RNA-seq library preparation and sequencing

For comparing 36–48 hpf bulk RNA-seq samples an RNEasy Micro Kit (QIAGEN, 74004) was used for RNA extraction of cell lysates from FAC-sorted cells. RNA quality was checked at the UC Irvine Genomics High Throughput Facility (GHTF; RRID:SCR_026615) using a Bioanalyzer 2100 (Agilent; RRID:SCR_018043). The Smart-seq2 protocol (RRID:SCR_021228) was utilized for cDNA library construction (*Picelli et al., 2014*). Libraries were sequenced at the GHTF using a HiSeq 4000 sequencer (Illumina; RRID:SCR_016386) at a read depth of ~35 M reads per replicate. From 11 total biological replicates (7 for 36 hpf, 4 for 48 hpf) we obtained approximately 10,000 cells per sample replicate.

For 48 hpf bulk RNA-seq experiments, library preparations from aBTX-injected and uninjected siblings were performed by the UCI GHTF. Libraries were sequenced at GHTF on a NextSeq 550 sequencer (Illumina; RRID:SCR_016381) at a read depth of ~35 M reads per replicate.

## Bulk RNA-seq data analysis

Bulk RNA-seq reads were mapped to the zebrafish genome version GRCz10 and quantified using STAR v2.5.2a (RRID:SCR_004463) (*Dobin et al., 2013*) and RSEM v1.2.31 (RRID:SCR_000262) (*Li and Dewey, 2011*). Differential gene expression analysis and PCA were performed using R package DESeq2 v1.30.1 (RRID:SCR_015687) (*Love et al., 2014*). Pairwise comparisons were performed between 36 and 48 hpf sorted tenocytes, and a Benjamini–Hochberg FDR adjusted p-value <0.05 was used as a threshold for considering significant differences in gene expression levels. PCA was performed on normalized count data which underwent variance-stabilization-transformation using DESeq2. Heatmaps were generated using ClustVis (RRID:SCR_017133) (*Metsalu and Vilo, 2015*). GO term enrichment analysis was performed using the ClusterProfiler R package (RRID:SCR_016884) (*Wu et al., 2021*) and ShinyGO (RRID:SCR_019213) (*Ge et al., 2020*).

## aBTX injections

aBTX mRNA was synthesized from the *pmtb-t7-alpha-bungarotoxin* vector (Megason lab, Addgene, 69542; RRID:Addgene_69542) as described in *Subramanian et al., 2018*; *Subramanian and Schilling, 2014* and injected into embryos at the 1-cell stage at a volume of 500 picoliters per embryo. aBTX mRNA was injected at a concentration of 90 ng/µl (45 pg/embryo)to paralyze embryos that were collected for analysis at 48 hpf and 150 ng/µl (90 pg/embryo) to paralyze embryos that were collected for analysis at 72 hpf.

## RT-qPCR

WT, *cacnb1*$^{-/-}$, aBTX-paralyzed, twitching, and recovered embryos were collected at respective time-points, homogenized in Trizol with prefilled tube kits using high impact zirconium beads (Benchmark Scientific, D1032-10) using a BeadBug 3 Microtube Homogenizer D1030 (Benchmark Scientific), and RNA was extracted as described previously (*Subramanian et al., 2018*). cDNA was prepared according to the standard oligo-dT primer protocol using the ProtoScript II First Strand cDNA Synthesis Kit (NEB E6560). cDNA was diluted 1:25 in water and used as template for RT-qPCR using the Luna Universal qPCR master mix (NEB M3003S). Primers used are listed in *Supplementary file 7*. Primer efficiencies were calculated with the formula PCR-efficiency = $10^{(-1/slope)}$ from a linear regression of Cp/ln(DNA) using a serial dilution of each primer with 72 hpf embryo cDNA as described in *Pfaffl, 2001*. PCR reactions were performed on a LightCycler 480 II Real Time PCR Instrument (Roche; RRID:SCR_018626) and analyzed using LightCycler 480 Software (Roche; RRID:SCR_012155). Each RT-qPCR experiment was repeated in triplicate for each biological replicate, and at least two biological replicates were used for each analysis. p-values were calculated using a two-tailed Student's *t*-test with $\alpha$ = 0.05 in Microsoft Excel. Bar charts in *Figure 3* present mean ± standard error. Venn diagram was created using the VennDiagram v1.7.3 (RRID:SCR_002414) R package with the gene list overlap tested with the Fisher's exact test from the GeneOverlap v1.26.0 (RRID:SCR_018419) R package (*Li Shen, 2017*).

## Imaging and *is*HCR quantification

Whole embryos imaged for chromogenic ISH were mounted on slides in 80% glycerol and imaged using a Zeiss Axioplan 2 compound microscope (RRID:SCR_020918) utilizing an AxioCam 305 Color Micropublisher 5.0 RTV camera with Zeiss Zen 3.1 (blue edition; RRID:SCR_013672) software. Embryos imaged for *is*HCR were embedded in 1% low melting point agarose/5× SSC and imaged on a Leica SP8 confocal microscope (RRID:SCR_018169) using the PL APO CS2 40×/1.10 W objective. Whole embryos imaged for *is*HCR were mounted in slide dishes in 1% low melt agarose with either 5× SSC (if only *is*HCR was performed) or 1× PBT (if *is*HCR combined with immunofluorescence was performed) and imaged using a Leica SP8 confocal microscope with LASX software (RRID:SCR_013673). *is*HCR voxel colocalizations in *Figure 2* and *Figure 2—figure supplement 1* were performed using the 'Coloc' function in Imaris 10.0.1 (RRID:SCR_007370) at the Optical Biology Core (RRID:SCR_026614) as described in *Subramanian et al., 2023*. Voxel colocalization only shows overlap of fluorescent channels within a particular voxel which may not, in some instances, fully reflect actual colocalization of fluorescence within a particular cell due to the punctate nature of *is*HCR fluorescence. *is*HCR single-cell quantification was performed in Imaris 10.0.1 using DAPI (Sigma D9542; PubChem CID: 2954) as a nuclear marker, as described in *Subramanian et al., 2023*. Embryo imaging for a single experiment was performed with identical parameters across conditions. Briefly, an ROI of the DAPI-stained nucleus from each 3D stack was traced through individual z-slices and mean voxel-intensity (AU) was measured. *matn1*/Scxa co-expressing cells measured were located at the ima enthesis on Meckels cartilage and sht enthesis at the anterior edge of the ch cartilage. *klf2a*/Scxa and *mxra5b*/Scxa co-expressing cells measured were located at the ima enthesis and sht enthesis, mhj MTJ and sht MTJ. Experimental conditions pertaining to each embryo image were saved separately, measurements were performed, and conditions were matched to each image. All p-values were calculated using a linear mixed effects model with individual embryos set as the random variable, and cells set as the fixed variable using the lme4 and lmetest R packages. Tukey–Kramer post hoc tests for pairwise analyses were then performed (ns = not significant, *p < 0.05, **p < 0.01, ***p < 0.001).

## Multiplex CRISPR–Cas9 genome editing of *matn1*, *klf2a*, and *mxra5b*

*matn1*, *klf2a*, and *mxra5b* multiplex gRNA injections were performed using the methodology described in *Wu et al., 2021* using gRNA primer sequences obtained from the primer database provided. Briefly, PCR was performed with four primers (per gene) targeting coding regions with T7 and spacer sequences for template gRNA synthesis. Transcription was performed with the T7 Megashortscript kit (Invitrogen AM1354). A 500-ng/μl solution of all four gRNAs were incubated at 37°C and injected into 1-cell stage embryos at a 500-pl volume per embryo.

## Acknowledgements

We would like to acknowledge Dr. Daniel Dranow for reviewing the manuscript and assistance provided for experimental design. This study was made possible in part through access to the Optical Biology Core Facility of the Developmental Biology Center, a shared resource supported by the Cancer Center Support Grant (CA-62203). This work was supported by the National Science Foundation (MCB2028424), the National Institutes of Health (R01 DE13828, R01 DE30565, and R01 AR67797 to TFS) and by a fellowship awarded to PKN from the National Science Foundation-Simons Center for Multiscale Cell Fate supported by the Simons Foundation (594598).

## Additional information

### Funding

| Funder | Grant reference number | Author |
| --- | --- | --- |
| National Science Foundation | MCB2028424 | Thomas F Schilling |
| National Institutes of Health | R01 DE13828 | Thomas F Schilling |
| National Institutes of Health | R01 DE30565 | Thomas F Schilling |
| National Institutes of Health | R01 AR67797 | Thomas F Schilling |
| National Science Foundation- Simons Center for Multiscale Cell Fate | 594598 | Pavan K Nayak |

The funders had no role in study design, data collection, and interpretation, or the decision to submit the work for publication.

### Author contributions

Pavan K Nayak, Conceptualization, Data curation, Formal analysis, Validation, Investigation, Visualization, Methodology, Writing - original draft, Writing – review and editing; Arul Subramanian, Conceptualization, Formal analysis, Validation, Visualization, Methodology, Writing – review and editing; Thomas F Schilling, Resources, Supervision, Funding acquisition, Investigation, Project administration, Writing – review and editing

### Author ORCIDs

Pavan K Nayak ⓘ https://orcid.org/0000-0002-4360-6729
Arul Subramanian ⓘ https://orcid.org/0000-0001-8455-6804
Thomas F Schilling ⓘ https://orcid.org/0000-0003-1798-8695

### Ethics

This study was performed in strict accordance with the recommendations in the Guide for the Care and Use of Laboratory Animals of the National Institutes of Health. All of the animals were handled according to approved Institutional Animal Care and Use Committee (IACUC) protocols (#AUP-23-099) of the University of California Irvine. The protocol was approved by the UCI IACUC Committee and ULAR (University Laboratory Animal Welfare).

### Decision letter and Author response

Decision letter https://doi.org/10.7554/eLife.105802.sa1
Author response https://doi.org/10.7554/eLife.105802.sa2

## Additional files

### Supplementary files

Supplementary file 1. Differentially expressed gene list of bulk RNA-seq of sorted mCherry+

tenocytes from 36 hpf vs 48 hpf embryos.

Supplementary file 2. ShinyGO analysis of 36 vs. 48 hpf bulk RNA-seq differentially expressed genes.

Supplementary file 3. DAVID analysis of 36 vs. 48 hpf bulk RNA-seq differentially expressed genes.

Supplementary file 4. Differentially expressed gene list from bulk RNA-seq of sorted mCherry+ tenocytes from 48 hpf WT vs aBTX-injected embryos.

Supplementary file 5. List of differentially expressed genes overlapping between 36 hpf vs 48 hpf bulk RNA-seq and 48 hpf WT vs a-BTX injected paralysis bulk RNA-seq.

Supplementary file 6. ShinyGO analysis of differentially expressed genes overlapping between 36 hpf vs 48 hpf bulk RNA-seq and 48 hpf WT vs a-BTX injected paralysis bulk RNA-seq.

Supplementary file 7. List of primers used for chromogenic in situ hybridizations and RT-qPCRs.

MDAR checklist

## Data availability

We have uploaded our datasets, software code, etc. to the GEO portal. We have received the GEO accession numbers for the datasets – GSE292682 and GSE292683. All source data (quantification data) have been uploaded with the manuscript and referred to in the figure legends, respectively.

The following datasets were generated:

| Author(s) | Year | Dataset title | Dataset URL | Database and Identifier |
|---|---|---|---|---|
| Nayak PK, Subramanian A, Schilling TF | 2025 | Raw reads for bulk RNAseq of FAC sorted tenocytes from 36 hpf vs. 48 hpf zebrafish embryos | http://www.ncbi.nlm.nih.gov/geo/query/acc.cgi?acc=GSE292682 | NCBI Gene Expression Omnibus, GSE292682 |
| Nayak PK, Subramanian A, Schilling TF | 2025 | Raw reads of bulk RNAseq from FAC sorted tenocytes of 48 hpf WT vs. aBTX injected paralyzed zebrafish embryos | http://www.ncbi.nlm.nih.gov/geo/query/acc.cgi?acc=GSE292683 | NCBI Gene Expression Omnibus, GSE292683 |

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
