## [Editor Report]

This valuable manuscript presents solid evidence that identifies potential force-responsive gene expression responses within tenocytes and developing tendons by comparing unperturbed animals to those with paralyzed muscles. A handful of these force-responsive genes are then validated, which reveals that force-responsive gene expression differs between individual tendons or local biophysical environments and shows a phenotype in mutants for a force-responsive gene expressed during tendon development. Future work that further explores how these particular examples relate to broader force-responsive gene expression programs and that identifies stronger phenotypes when force-responsive gene expression is disrupted will strengthen its conclusions. This work is of interest to the fields of developmental biology, mechanobiology, muscle and tendon biology.

---

## [Decision Letter]

**Decision letter after peer review:**

Thank you for submitting the paper "Transcriptome profiling of tendon fibroblasts at the onset of embryonic muscle contraction reveals novel force-responsive genes" for consideration by *eLife*. Your article has been reviewed by 3 peer reviewers who have opted to remain anonymous.

Comments to the Authors:

We are sorry to say that the reviewers have decided that this work will not be considered further for publication by *eLife*. The manuscript has potential but our view is that substantial and possibly repeated revisions would be needed before it could be accepted.

*Reviewer #1 (Recommendations for the authors):*

The authors exploit an important transition in zebrafish development to understand force response in tenocytes. They identify over 1000 genes that respond to force (by muscle contraction), providing an extensive list of genes to evaluate for roles in mechanoresponse and tendon maturation. While the experiments presented in this manuscript initiate an exciting research program that will advance our understanding of tendon biology, the limited focus on three genes with distinct expression profiles and responses to force limit the interpretation of the presented data. The observation of discrete expression profiles for tendons across the skeleton is interesting, but more work will be necessary to establish the principles of general and local tenocyte response to force. This work continues an exciting direction of study for the authors that will likely reveal tendon-specific and general mechanisms of force response.

The basis of the manuscript centers on changes driven by the muscle contraction that initiates between 36 hpf and 48 hpf, an event that the authors have previously demonstrated drives a range of interesting morphological changes in tendon cells. The authors capture several genes that change gene expression as muscle contraction initiates, including three factors (klf2a, matn1, and mxra5b) that they focus on based on previous literature implicating them with known mechano-responsive pathways. However, whether these genes are direct responders to force or further downstream responders that control the morphological changes previously described is unclear. Despite possible issues with redundancy, the authors should provide a characterization of stable mutant lines from the crispants described in the discussion to strengthen the manuscript.

While the authors discuss possible pathways upstream of candidate genes (ie TGF-β upstream of matn1), the authors do not test the responsiveness of these genes to the predicted pathways in zebrafish, which reduces the impact of relating published studies.

The discrepancies between the experiments that attempt to validate gene expression changes captured in their RNA-seq experiment make the ultimate conclusions of this paper hard to interpret. While the authors suggest interesting ideas about the reasoning for each specific inconsistency, the discrepancies make it unclear which experiment reflects the biological processes described. For example, although not explicitly stated, Figure 4C suggests mxra5b is down-regulated between 36 and 48 hpf, suggesting it is down-regulated as the muscles contract. This is in contrast with the results from the RT-qPCR experiments and in situ-based experiments, confounding conclusions. The authors' results with klf2a are especially hard to interpret, as each experiment shows a different result. Unless the authors provide additional genes with similar responses, the data weakens the manuscript.

Given that their experiments are focused on purified tendon cells, the authors should consider removing all whole animal RT-qPCR experiments from the manuscript. In exchange, fluorescent-based experiments using tendon markers should be repeated in cacnb1-/- fish.

In addition, because the authors only focus on three genes that happen to have different expression patterns with tendons and different responses to force, the hypothesized model for the diverse local responses is poorly supported. Expanding their study to include genes that share similar gene expression profiles and responses to force will strengthen the study.

The authors should consider including RNA-seq data from cacnb1-/- and aBTX-P treated fish to expand their list of relevant force-responsive genes with the patterns captured from their analysis. Additionally, the authors suggest divergent force-responsive features between the trunk and craniofacial tendons. To support this beyond the description of three genes, the authors should consider performing RNA-seq experiments that compare force response between craniofacial and trunk tendons.

*Reviewer #2 (Recommendations for the authors):*

This manuscript addresses the useful question of how movement impacts tendon development by taking the approach of transcriptomic analysis of tendon precursor cells before and after swimming. The authors claim that they identified novel tendon-associated genes, three of which were investigated further by conducting paralysis/rescue studies to show that expression is force-responsive.

One weakness of this manuscript is that it is not well grounded in extant literature regarding the development and differentiation of cartilage/tendons, but more importantly with previous deep sequencing of scx-expressing cells (mESC-derived, mammalian, porcine, and meta-analyses have been conducted – for example, PMIDs PMC8270956, PMC5037390, PMC7810463, PMC5751986). The manuscript is quite thin, and one approach to filling the manuscript out could be to significantly increase the integration and analysis of their data that focuses on changes occurring during development.

The authors rest a lot of the significance of the study on the novelty of the genes identified, but this is also somewhat confusing, "By paralyzing and restoring muscle contractions in embryos in vivo, we show that three of these novel genes….). For example, mxra5b does not appear to be novel (identified in avian development as expressed in tendons/ligaments (PMID 29877573) and in adult pig tendons (PMC5751986). Similarly, kruppel-like factors have been identified in both being force responsive and as regulators of bi-fated tendon-to-bone attachment cells.

The data methodology/interpretation is either confusing or this reviewer is confused, but the data do not necessarily seem to agree with the stated conclusions. In general, this is a potentially exciting manuscript but, in its current form, is not presented as rigorous or compelling enough.

Example 1 Figure 3 – sxca ish is a little bit difficult to believe as the red dots look like they could be background as well, could the authors cite previous clear data showing that the unimpressive red dots are real? Also, could the authors discuss the other attachment sites (presumably there are some) where co-expression was not observed, and discuss the implications of these data? This figure is very difficult for the reader to scan – it would be helpful to include more labels/graphics, etc. Also – is this a one-off or was this observed in multiple embryos? And, if swimming has already started by 72hpf, was ish done prior to this to show co-expression? It seems like showing expression after swimming is contrary to the point note the figure says 72hpf the text says 51hpf but the figure looks more like the older fish could look in figure 2 – a bit difficult because orientation seems to be different.

Also, co-expression by ish of kfl2a and scxa is difficult in this figure because it is not clear what the white outlines correspond to – they don't clearly overlap with dapi, thus how are they drawn? How frequently was this observed – in all somite boundaries? Similar questions abound with mxra5b – this seems to have such low expression, is it all co-localized with scxa, and what happens later in development?

Example 2 Figure 4 – could the authors please speak to why matn1 is not significantly downregulated at attachment sites in paralyzed embryos versus wild-type in panel I but it is significantly downregulated in whole embryos in panel E? Does this data suggest that matn1 is not force-responsive at attachment sites? Can the authors please explain how the swimming force in the trunk impacts cranial cartilage development? That is one confusing aspect of this manuscript, how head cartilage development relates to swimming. Perhaps showing images of how cartilage and trunk tendons are affected in paralyzed embryos would be helpful.

*Reviewer #3 (Recommendations for the authors):*

In this work, the authors aim to delineate a model in which mechanical force during embryonic muscle development regulates novel genes in tendon fibroblasts. The role of mechanical forces in regulating gene programs is an exciting area of development and is leading to many unexpected findings across fields.

There are additional experiments that could significantly strengthen the appended manuscript:

1) It would help the reader to understand how clean the FACS-sorted cell population used for sequencing analysis is. In brief, it is hard to fully digest the genomics data without having an understanding of this tendon fibroblast population is pure.

2) Are there phenotypic effects on tendon development from loss of function of any of the identified genes? This remains a large open-ended question of this study.

3) There are instances where the authors use the word 'specific' for the expression patterns of these genes. This is a bit of a misnomer, as genes such as Klf2 are expressed in the vasculature and mxra5b in the notochord. This expression in alternative tissues speaks to why readers need to fully understand how clean and specific the isolated cell population used for sequencing was.

4) The downstream qPCR analysis from this study is done globally on whole fish embryos. Given that these genes are not specific, it is difficult to relate changes in gene expression in whole fish to specific cell types.

5) The authors' use of fluorescent in situ hybridization to increase resolution and assess the overlap of their genes of interest with known genes expressed in tendon fibroblasts is commendable. Unfortunately, most of the expression overlap of interest is on the order of pixel-level resolution and is difficult to interpret for a broad audience.

The authors should be commended for an attempt to address this critical area of biology, however, the manuscript in its current form is lacking the data and resolution to fully support their conclusions.

While the concept presented in this manuscript is extremely interesting and could be of high value to the field, as things currently stand, there isn't enough supporting evidence to support the publication of this work.

Specific suggestions that would need to be addressed:

1) the isolated cell population used for sequencing needs to be validated. The authors need to confirm that there are no contaminating cells, such as endothelial cells or notochord cells, in their preps. Without this, much of the data could be artifactual. For instance, in Figure 3F, almost the entirety of klf2a expression is in the vasculature. While the authors highlight potential areas of interest, it is impossible to tell from confocal stacks if this is a true overlap of expression or just expression in different planes that appear overlapping because the images are Z-projected. In general, the images are not particularly compelling to hinge on the entirety of the manuscript on.

2) Given comment 1, the paper could be strengthened by demonstrating the functional effect of these genes in tendon fibroblasts. What happens if they aren't there? Are there consequences for tendon development? Without this, it is hard to fully realize the implications of the authors' data.

3) While Figure 4 shows that there could be a link between mechanics and expression of these genes, the model used to test this relies on an ion channel loss of function setting. It is nearly impossible to decouple the change in gene expression from the loss of mechanics in this model from the loss of the gating current of the channel. For this reason, additional orthogonal models should be used to bolster and support the authors' claims.

These points should be addressed before a deeper review of the conclusions from this manuscript could be fully addressed.

---

## [Author Response]

We thank the reviewers for their constructive comments. Here we list our point-by-point responses and have revised the manuscript accordingly.

Multiple reviewers points.

1. All 3 reviewers asked for a better explanation of the changes in *matn1, klf2a,* and *mxra5b* expression in the bulk RNA-seq experiments with *cacnb1*^-/-^ mutants to clarify if they were due to paralysis or instead to gene expression changes reflecting embryonic tendon maturation. To address this, we conducted an additional bulk RNA-seq experiment at 48 hpf (hours postfertilization) on sorted mCherry+ tenocytes from *α bungarotoxin* (aBTX) injected Tg(*scxa:mCherry*) paralyzed embryos and uninjected controls. Changes in *matn1, klf2a,* and *mxra5b* expression showed a similar trend to our bulk RNA-seq results at 36-48 hpf (Figure 3G), supporting the hypothesis that they are caused by changes muscle contractile forces. We have also added discussion of how these gene expression changes differ from those associated with tenocyte maturation.

2. All 3 reviewers were concerned about contamination of sorted tenocytes with other cell types, in particular because the RT-qPCR results with sorted cells differed from those found by bulk RNA-seq. To address this we provide data showing the stringency of our FACS gating thresholds for the mCherry+ signal (Figure 1—figure supplement 1). In addition, tenocyte-specific expression profiles of *matn1, klf2a,* and *mxra5b* were quantified using IMARIS 3D imaging software to select individual tenocyte nuclei from HCRish’s at two different tendon attachments for *matn1* and four different attachments for *klf2a*, and *mxra5b,* across four different contractile force conditions (Figures 2, 4, 5, 6 and all supplements for these figures). These data reveal not only temporal but also spatial heterogeneity in expression of *matn1, klf2a,* and *mxra5b* in tenocyte subpopulations in response to force.

3. Several reviewers criticized figure quality, particularly for our HCRish analyses. Former figures 2 and 3 (current figures 2, 4, 5, 6 and all associated supplements) have now been completely revised to include high resolution images of HCRishs for *matn1*, *mxra5b*, and *klf2a.* These show co-localization with *scxa* using IMARIS software. The original chromogenic ISH images have been moved to the supplements (See Figure 2—figure supplement 1).

4. Several reviewers suggested we remove the RT-qPCR results. However, these were used initially to validate genes with mechanosensitive activation in our bulk RNA-seq experiments. Global expression changes occur in tissues other than tenocytes (e.g. *matn1* in cartilage, *klf2a* in vasculature) in these embryos. Going by the concerns of the reviewers, we have decided to move the RT-qPCR figure to supplementary data (Figure 3—figure supplement 1). Our HCRish measurements show spatially restricted expression to tenocytes and how they are affected by varying force.

Reviewer #1 (Recommendations for the authors):The authors exploit an important transition in zebrafish development to understand force response in tenocytes. They identify over 1000 genes that respond to force (by muscle contraction), providing an extensive list of genes to evaluate for roles in mechanoresponse and tendon maturation. While the experiments presented in this manuscript initiate an exciting research program that will advance our understanding of tendon biology, the limited focus on three genes with distinct expression profiles and responses to force limit the interpretation of the presented data. The observation of discrete expression profiles for tendons across the skeleton is interesting, but more work will be necessary to establish the principles of general and local tenocyte response to force. This work continues an exciting direction of study for the authors that will likely reveal tendon-specific and general mechanisms of force response.

See response to multiple reviewers #1 above

1. The basis of the manuscript centers on changes driven by the muscle contraction that initiates between 36 hpf and 48 hpf, an event that the authors have previously demonstrated drives a range of interesting morphological changes in tendon cells. The authors capture several genes that change gene expression as muscle contraction initiates, including three factors (klf2a, matn1, and mxra5b) that they focus on based on previous literature implicating them with known mechano-responsive pathways. However, whether these genes are direct responders to force or further downstream responders that control the morphological changes previously described is unclear. Despite possible issues with redundancy, the authors should provide a characterization of stable mutant lines from the crispants described in the discussion to strengthen the manuscript.

All 3 reviewers asked for a better explanation of the changes in *matn1, klf2a,* and *mxra5b* expression in the bulk RNA-seq experiments with *cacnb1*^-/-^ mutants to clarify if they were due to paralysis or instead to gene expression changes reflecting embryonic tendon maturation. To address this, we conducted an additional bulk RNA-seq experiment at 48 hpf (hours postfertilization) on sorted mCherry+ tenocytes from *α bungarotoxin* (aBTX) injected Tg(*scxa:mCherry*) paralyzed embryos and uninjected controls. Changes in *matn1, klf2a,* and *mxra5b* expression showed a similar trend to our bulk RNA-seq results at 36-48 hpf (Figure 3G), supporting the hypothesis that they are caused by changes muscle contractile forces. We have also added discussion of how these gene expression changes differ from those associated with tenocyte maturation.

2. While the authors discuss possible pathways upstream of candidate genes (ie TGF-β upstream of matn1), the authors do not test the responsiveness of these genes to the predicted pathways in zebrafish, which reduces the impact of relating published studies.

Our study focuses on force responses of *klf2a, matn1, and mxra5b*, depending on the tendon type and interface (e.g. MTJ or enthesis). From published literature (including work from our lab) we know that TGFb and other mechanotransduction pathways regulate these responses in tendon tissue. Our work focuses on the mechanosensitive activation of genes during a key developmental stage marking the transition from sporadic movements to free-swimming behavior. Comprehensive analysis of various mechanotransduction pathways regulating each gene is beyond the scope of this paper.

3. The discrepancies between the experiments that attempt to validate gene expression changes captured in their RNA-seq experiment make the ultimate conclusions of this paper hard to interpret. While the authors suggest interesting ideas about the reasoning for each specific inconsistency, the discrepancies make it unclear which experiment reflects the biological processes described. For example, although not explicitly stated, Figure 4C suggests mxra5b is down-regulated between 36 and 48 hpf, suggesting it is down-regulated as the muscles contract. This is in contrast with the results from the RT-qPCR experiments and in situ-based experiments, confounding conclusions. The authors' results with klf2a are especially hard to interpret, as each experiment shows a different result. Unless the authors provide additional genes with similar responses, the data weakens the manuscript.

See also the response to multiple reviewers #1 above. We now include spatial expression data in addition to the temporal gene expression data (See Figure 2, 4, 5, 6, and all associated supplements). We propose a model in which *matn1, klf2a,* and *mxra5b* respond to force differently depending on tissue attachment type (i.e. hard vs. soft; enthesis vs MTJ), force intensity, and persistence (paralyzed vs twitching vs fully recovered) (See Figure 8).

4. Given that their experiments are focused on purified tendon cells, the authors should consider removing all whole animal RT-qPCR experiments from the manuscript. In exchange, fluorescent-based experiments using tendon markers should be repeated in cacnb1-/- fish.

See response to multiple reviewers #4 above. Our spatial analyses of gene expression now include HCRish in different tendons, entheses and MTJs in aBTX-injected embryos rather than *cacnb1-/-*. In previous work in our lab, we know that both aBTX and *cacnb1-/-* show similar phenotypes and effects on tenocyte development (Subramanian et al. 2018, 2023).

6. In addition, because the authors only focus on three genes that happen to have different expression patterns with tendons and different responses to force, the hypothesized model for the diverse local responses is poorly supported. Expanding their study to include genes that share similar gene expression profiles and responses to force will strengthen the study.

We focus on genes differentially expressed between two key developmental time points during which there is a dramatic change in intensity and persistence of force from muscle activity. We chose *matn1*, *mxra5b*, and *klf2a* based on previous studies supporting a role for force in their expression in other tissues and little evidence for their roles for mechanotransduction in tendons. We utilized the in-depth analysis of these genes to establish a new paradigm of mechanosensitive activation (i.e. that it is tissue specific and not occurring necessarily in a binary on/off modality). Including other genes into this study may show different variability of expression, which is beyond the scope of this paper.

6. The authors should consider including RNA-seq data from cacnb1-/- and aBTX-P treated fish to expand their list of relevant force-responsive genes with the patterns captured from their analysis. Additionally, the authors suggest divergent force-responsive features between the trunk and craniofacial tendons. To support this beyond the description of three genes, the authors should consider performing RNA-seq experiments that compare force response between craniofacial and trunk tendons.

See also our response to point #1 above. Paralysis induced by *cacnb1^-/-^* or via injection of aBTX have similar effects on tenocyte morphogenesis (Subramanian et al. 2018) and gene expression (Subramanian et al. 2023). Many of the force-responsive genes identified overlap between the two RNA-seq experiments (See Figure 3E) and we find similar temporal and spatial changes in *matn1*, *mxra5b*, and *klf2a* expression in both datasets (See Figure 3D).

Reviewer #2 (Recommendations for the authors):1. This manuscript addresses the useful question of how movement impacts tendon development by taking the approach of transcriptomic analysis of tendon precursor cells before and after swimming. The authors claim that they identified novel tendon-associated genes, three of which were investigated further by conducting paralysis/rescue studies to show that expression is force-responsive.One weakness of this manuscript is that it is not well grounded in extant literature regarding the development and differentiation of cartilage/tendons, but more importantly with previous deep sequencing of scx-expressing cells (mESC-derived, mammalian, porcine, and meta-analyses have been conducted – for example, PMIDs PMC8270956, PMC5037390, PMC7810463, PMC5751986). The manuscript is quite thin, and one approach to filling the manuscript out could be to significantly increase the integration and analysis of their data that focuses on changes occurring during development.

We thank the reviewer for pointing out these studies, which are largely in vitro using mammalian cells and difficult to compare with our in vivo approaches in zebrafish development. PMC7810463 shows expression of KLF2 during murine enthesis development, but does not investigate effects of force on KLF2 expression. PMC5751986 compares KLF2 expression between tendon and enthesis but again not in the context of force.

2. The authors rest a lot of the significance of the study on the novelty of the genes identified, but this is also somewhat confusing, "By paralyzsing and restoring muscle contractions in embryos in vivo, we show that three of these novel genes….). For example, mxra5b does not appear to be novel (identified in avian development as expressed in tendons/ligaments (PMID 29877573) and in adult pig tendons (PMC5751986). Similarly, kruppel-like factors have been identified in both being force responsive and as regulators of bi-fated tendon-to-bone attachment cells.

Our study shows the first evidence of a role for mechanical force from muscle contraction on *mxra5b, klf2a* or *matn1* expression. Since some previous work has implicated these genes in force responses and/or in tendons, we have largely avoided use of “novel” in the manuscript when referring to *matn1, klf2a,* and *mxra5b*.

3. The data methodology/interpretation is either confusing or this reviewer is confused, but the data do not necessarily seem to agree with the stated conclusions. In general, this is a potentially exciting manuscript but, in its current form, is not presented as rigorous or compelling enough.Example 1 Figure 3 – sxca ish is a little bit difficult to believe as the red dots look like they could be background as well, could the authors cite previous clear data showing that the unimpressive red dots are real? Also, could the authors discuss the other attachment sites (presumably there are some) where co-expression was not observed, and discuss the implications of these data? This figure is very difficult for the reader to scan – it would be helpful to include more labels/graphics, etc. Also – is this a one-off or was this observed in multiple embryos? And, if swimming has already started by 72hpf, was ish done prior to this to show co-expression? It seems like showing expression after swimming is contrary to the point note the figure says 72hpf the text says 51hpf but the figure looks more like the older fish could look in figure 2 – a bit difficult because orientation seems to be different.

Please see our response to multiple reviewers #1 and #3. Revised figures show HCRish images of *matn1, klf2a*, and *mxra5b* expression, including close-up views of selected dual *scxa* expressing nuclei, colocalizing voxels with IMARIS software.

1. Figure 1 now illustrates the embryonic zebrafish axial and cranial musculoskeletal systems including the relevant tendons in this study, also labeled in other figures.

2. Revised HCRish data were reanalyzed in IMARIS for colocalization with scxa. It is true that some attachment sites lack *scxa* tenocytes, (1) e.g. *matn1* expressing tenocytes in the mandibulohyoid junction (mhj) tendon as this is a muscle-muscle attachment, as well as the sternyhoideus (sh) MTJ since there is no adjacent cartilage. *matn1* is primarily expressed in cartilages and entheses as we show in the new Figure 2, Figure 4, and associated supplements.

3. Co-localization with *scxa* was performed and quantified for *matn1, klf2a*, and *mxra5b* in HCRish with multiple embryos, but we include one representative image for each (See Figure 2, 4, 5, 6 and all associated supplements). We have also expanded the Methods section describing in detail numbers of embryos imaged and tenocytes quantified for each tendon. Including expression changes after onset of swimming behavior is relevant, especially for *matn1* and *klf2a,* since they are upregulated upon initiation of muscle contraction, as we showed in the original submission with bulk RNA-seq.

As described in our response to multiple reviewers #1, to confirm roles for force in regulation of *matn1*, *klf2a,* and *mxra5b* expression we have repeated paralysis experiments using aBTX-injection, with similar results. Over 200 genes are shared between the two datasets suggesting that onset of force rather than simply tendon maturation account for these results.

4. Also, co-expression by ish of kfl2a and scxa is difficult in this figure because it is not clear what the white outlines correspond to – they don't clearly overlap with dapi, thus how are they drawn? How frequently was this observed – in all somite boundaries? Similar questions abound with mxra5b – this seems to have such low expression, is it all co-localized with scxa, and what happens later in development?

We have used IMARIS to improve the HCRish images dramatically. We have revised the images and now present the co-expression of *klf2a* and *scxa* in both the trunk at 48 hpf and the head at 72 hpf along with magnified views of individual nuclei which contain colocalization of *klf2a* and *scxa* fluorescently labelled transcripts using the voxel colocalization tool available in IMARIS software (See Figure 2G-P). Co-localization in the trunk is observed at most somite boundaries. *mxra5b* expression in the trunk does not obviously colocalize with scxa, though it localizes to somite boundaries (See Figure 2G-I). Cranial *mxra5b* and *scxa* clearly co-localize in cranial tenocytes (See Figure 2Q-V).

5. Example 2 Figure 4 – could the authors please speak to why matn1 is not significantly downregulated at attachment sites in paralyzed embryos versus wild-type in panel I but it is significantly downregulated in whole embryos in panel E? Does this data suggest that matn1 is not force-responsive at attachment sites? Can the authors please explain how the swimming force in the trunk impacts cranial cartilage development? That is one confusing aspect of this manuscript, how head cartilage development relates to swimming. Perhaps showing images of how cartilage and trunk tendons are affected in paralyzed embryos would be helpful.

Please see our responses to multiple reviewers #1 and #3. We have revised Figure 3 and moved RT-qPCR results to Figure 3- supplementary figure 1. Differences between the whole embryo RT-qPCR and HCRish results can be explained by the nature of the measurements. *matn1* expression shows no expression in the trunk tenocytes, but shows changes in expression between 36 hpf and 48 hpf, and between WT and paralyzed embryos at 48hpf in the bulk RNAseq. This can be because: 1) *matn1* expression is also localized to the pec fin buds (which are not part of the head tendons), which are contracting during the onset of swimming behavior and 2) at 48 hpf, *matn1* expression also localizes to the developing jaw cartilages and tendons (See Figure 1C and Figure 2. Supplement 2), which lie ventrally closer to the trunk muscles and heart before extending anteriorly by 60 hpf. Therefore it is possible that the forces from contractions of the trunk musculature and pec fins have an effect on *matn1* expression in the cranial tissues as well indeed, we see mechanosensitive transcriptional patterns of *matn1* expression in the RT-qPCRs, and new bulk RNAseq involving aBTX injected paralyzed embryos at 48 hpf, see Figure 3A and G).

Reviewer #3 (Recommendations for the authors):In this work, the authors aim to delineate a model in which mechanical force during embryonic muscle development regulates novel genes in tendon fibroblasts. The role of mechanical forces in regulating gene programs is an exciting area of development and is leading to many unexpected findings across fields.There are additional experiments that could significantly strengthen the appended manuscript:1) It would help the reader to understand how clean the FACS-sorted cell population used for sequencing analysis is. In brief, it is hard to fully digest the genomics data without having an understanding of this tendon fibroblast population is pure.

Please see our response to multiple reviewers #2.

2) Are there phenotypic effects on tendon development from loss of function of any of the identified genes? This remains a large open-ended question of this study.

We performed multiplex CRISPR injections and saw no noticeable embryonic zebrafish phenotypes for *matn1* and *klf2a. mxra5b^-/-^* CRISPant embryos showed disruptions of somite boundary morphology as well as fewer scxa+ trunk tenocytes, which we have included as Figure 7. *matn1^-/-^* knockout mice show either no (Li et al. 2020), or very mild skeletal abnormalities and reduced collagen 2 fibril thickness and collagen 2 synthesis (Huang et al. 1999, Chen et al. 2016). We created a stable matn1 mutant line but mutants were viable and fertile with no obvious phenotype, though we will further pursue the possibility that these fish have mild cartilage/skeletal phenotypes as adults.

3) There are instances where the authors use the word 'specific' for the expression patterns of these genes. This is a bit of a misnomer, as genes such as Klf2 are expressed in the vasculature and mxra5b in the notochord. This expression in alternative tissues speaks to why readers need to fully understand how clean and specific the isolated cell population used for sequencing was.

Please see our response to multiple reviewers #2 and #3. New bulk-RNA-seq experiments combined with spatial gene expression data should alleviate concerns about tissue or cell-type specificity.

4) The downstream qPCR analysis from this study is done globally on whole fish embryos. Given that these genes are not specific, it is difficult to relate changes in gene expression in whole fish to specific cell types.

Please see our response to multiple reviewers #4 above

5) The authors' use of fluorescent in situ hybridization to increase resolution and assess the overlap of their genes of interest with known genes expressed in tendon fibroblasts is commendable. Unfortunately, most of the expression overlap of interest is on the order of pixel-level resolution and is difficult to interpret for a broad audience.

Please see our response to multiple reviewers #3. We have completely revised the HCRish images and quantification of the spatial expression data.

While Figure 4 shows that there could be a link between mechanics and expression of these genes, the model used to test this relies on an ion channel loss of function setting. It is nearly impossible to decouple the change in gene expression from the loss of mechanics in this model from the loss of the gating current of the channel. For this reason, additional orthogonal models should be used to bolster and support the authors' claims.

Please see our response to multiple reviewers #1. Both cacnb1-/- and a-BTX injection cause paralysis by different mechanisms. Both have been extensively used in zebrafish to cause paralysis and cause no obvious changes in embryogenesis or cranial musculoskeletal development until 5 dpf (Subramanian et al. 2018, Subramanian et al. 2023) and have been used extensively in other systems (Sohal et al. 1979).

References

Chen Y, Cossman J, Jayasuriya CT, Li X, Guan Y, Fonseca V, Yang K, Charbonneau C, Yu H, Kanbe K, Ma P, Darling E, Chen Q. Deficient Mechanical Activation of Anabolic Transcripts and Post-Traumatic Cartilage Degeneration in Matrilin-1 Knockout Mice. PLoS One. 2016 Jun 7;11(6):e0156676. doi: 10.1371/journal.pone.0156676. PMID: 27270603; PMCID: PMC4896629.

Eliasson P, Andersson T, Aspenberg P. Rat Achilles tendon healing: mechanical loading and gene expression. J Appl Physiol (1985). 2009 Aug;107(2):399-407. doi: 10.1152/japplphysiol.91563.2008. Epub 2009 Jun 18. PMID: 19541731.

G.S. Sohal, T.L. Creazzo, T.G. Oblak, Effects of chronic paralysis with α-bungarotoxin on development of innervation, Experimental Neurology, Volume 66, Issue 3, 1979, Pages 619-628, ISSN 0014-4886,

Guerquin MJ, Charvet B, Nourissat G, Havis E, Ronsin O, Bonnin MA, Ruggiu M, Olivera-Martinez I, Robert N, Lu Y, Kadler KE, Baumberger T, Doursounian L, Berenbaum F, Duprez D. Transcrsiption factor EGR1 directs tendon differentiation and promotes tendon repair. J Clin Invest. 2013 Aug;123(8):3564-76. doi: 10.1172/JCI67521. Epub 2013 Jul 25. PMID: 23863709; PMCID: PMC4011025.

Huang X, Birk DE, Goetinck PF. Mice lacking matrilin-1 (cartilage matrix protein) have alterations in type II collagen fibrillogenesis and fibril organization. Dev Dyn. 1999 Dec;216(4-5):434-41. doi: 10.1002/(SICI)1097-0177(199912)216:4/5<434::AID-DVDY11>3.0.CO;2-X. PMID: 10633862.

Huanhuan Liu, Can Zhang, Shouan Zhu, Ping Lu, Ting Zhu, Xiaonan Gong, Ziwang Zhang, Jiajie Hu, Zi Yin, Boon Chin Heng, Xiao Chen, Hong Wei Ouyang, Mohawk Promotes the Tenogenesis of Mesenchymal Stem Cells Through Activation of the TGFβ Signaling Pathway, *Stem Cells*, Volume 33, Issue 2, February 2015, Pages 443–455, https://doi.org/10.1002/stem.1866

Ito Y, Toriuchi N, Yoshitaka T, Ueno-Kudoh H, Sato T, Yokoyama S, Nishida K, Akimoto T, Takahashi M, Miyaki S, Asahara H. The Mohawk homeobox gene is a critical regulator of tendon differentiation. Proc Natl Acad Sci U S A. 2010 Jun 8;107(23):10538-42. doi: 10.1073/pnas.1000525107. Epub 2010 May 24. PMID: 20498044; PMCID: PMC2890854.

Kimura W, Machii M, Xue X, Sultana N, Hikosaka K, Sharkar MT, Uezato T, Matsuda M, Koseki H, Miura N. Irxl1 mutant mice show reduced tendon differentiation and no patterning defects in musculoskeletal system development. Genesis. 2011 Jan;49(1):2-9. doi: 10.1002/dvg.20688. Epub 2010 Dec 22. PMID: 21254332.

Lejard V, Blais F, Guerquin MJ, Bonnet A, Bonnin MA, Havis E, Malbouyres M, Bidaud CB, Maro G, Gilardi-Hebenstreit P, Rossert J, Ruggiero F, Duprez D. EGR1 and EGR2 involvement in vertebrate tendon differentiation. J Biol Chem. 2011 Feb 18;286(7):5855-67. doi: 10.1074/jbc.M110.153106. Epub 2010 Dec 20. PMID: 21173153; PMCID: PMC3037698.

Li P, Fleischhauer L, Nicolae C, Prein C, Farkas Z, Saller MM, Prall WC, Wagener R, Heilig J, Niehoff A, Clausen-Schaumann H, Alberton P, Aszodi A. Mice Lacking the Matrilin Family of Extracellular Matrix Proteins Develop Mild Skeletal Abnormalities and Are Susceptible to Age-Associated Osteoarthritis. Int J Mol Sci. 2020 Jan 19;21(2):666. doi: 10.3390/ijms21020666. PMID: 31963938; PMCID: PMC7013758.

Liu H, Zhang C, Zhu S, Lu P, Zhu T, Gong X, Zhang Z, Hu J, Yin Z, Heng BC, Chen X, Ouyang HW. Mohawk promotes the tenogenesis of mesenchymal stem cells through activation of the TGFβ signaling pathway. Stem Cells. 2015 Feb;33(2):443-55. doi: 10.1002/stem.1866. PMID: 25332192.

Maeda T, Sakabe T, Sunaga A, Sakai K, Rivera AL, Keene DR, Sasaki T, Stavnezer E, Iannotti J, Schweitzer R, Ilic D, Baskaran H, Sakai T. Conversion of mechanical force into TGF-β-mediated biochemical signals. Curr Biol. 2011 Jun 7;21(11):933-41. doi: 10.1016/j.cub.2011.04.007. Epub 2011 May 19. PMID: 21600772; PMCID: PMC3118584.

Nakamichi R, Ma S, Nonoyama T, Chiba T, Kurimoto R, Ohzono H, Olmer M, Shukunami C, Fuku N, Wang G, Morrison E, Pitsiladis YP, Ozaki T, D'Lima D, Lotz M, Patapoutian A, Asahara H. The mechanosensitive ion channel PIEZO1 is expressed in tendons and regulates physical performance. Sci Transl Med. 2022 Jun;14(647):eabj5557. doi: 10.1126/scitranslmed.abj5557. Epub 2022 Jun 1. PMID: 35648809.

Otabe K, Nakahara H, Hasegawa A, Matsukawa T, Ayabe F, Onizuka N, Inui M, Takada S, Ito Y, Sekiya I, Muneta T, Lotz M, Asahara H. Transcription factor Mohawk controls tenogenic differentiation of bone marrow mesenchymal stem cells in vitro and in vivo. J Orthop Res. 2015 Jan;33(1):1-8. doi: 10.1002/jor.22750. Epub 2014 Oct 13. PMID: 25312837; PMCID: PMC4294629.

Passini FS, Jaeger PK, Saab AS, Hanlon S, Chittim NA, Arlt MJ, Ferrari KD, Haenni D, Caprara S, Bollhalder M, Niederöst B, Horvath AN, Götschi T, Ma S, Passini-Tall B, Fucentese SF, Blache U, Silván U, Weber B, Silbernagel KG, Snedeker JG. Shear-stress sensing by PIEZO1 regulates tendon stiffness in rodents and influences jumping performance in humans. Nat Biomed Eng. 2021 Dec;5(12):1457-1471. doi: 10.1038/s41551-021-00716-x. Epub 2021 May 24. PMID: 34031557; PMCID: PMC7612848.

Subramanian A, Kanzaki LF, Galloway JL, Schilling TF. Mechanical force regulates tendon extracellular matrix organization and tenocyte morphogenesis through TGFbeta signaling. *ELife*. 2018 Nov 26;7:e38069. doi: 10.7554/*eLife*.38069. PMID: 30475205; PMCID: PMC6345564.

Subramanian A, Kanzaki LF, Schilling TF. Mechanical force regulates *Sox9* expression at the developing enthesis. Development. 2023 Aug 15;150(16):dev201141. doi: 10.1242/dev.201141. Epub 2023 Aug 18. PMID: 37497608; PMCID: PMC10445799.

Zhang J, Wang JH. The effects of mechanical loading on tendons--an in vivo and in vitro model study. PLoS One. 2013 Aug 19;8(8):e71740. doi: 10.1371/journal.pone.0071740. PMID: 23977130; PMCID: PMC3747237.